# Selective filling of n-hexane in a tight nanopore

Haoran Qu [1], Archith Rayabharam [2], Xiaojian Wu [1], Peng Wang [1], Yunfeng Li[1], Jeffrey Fagan [3], Narayana R. Aluru [2] & YuHuang Wang [1,4✉]

Molecular sieving may occur when two molecules compete for a nanopore. In nearly all known examples, the nanopore is larger than the molecule that selectively enters the pore. Here, we experimentally demonstrate the ability of single-wall carbon nanotubes with a van der Waals pore size of 0.42 nm to separate n-hexane from cyclohexane—despite the fact that both molecules have kinetic diameters larger than the rigid nanopore. This unexpected finding challenges our current understanding of nanopore selectivity and how molecules may enter a tight channel. Ab initio molecular dynamics simulations reveal that n-hexane molecules stretch by nearly 11.2% inside the nanotube pore. Although at a relatively low probability (28.5% overall), the stretched state of n-hexane does exist in the bulk solution, allowing the molecule to enter the tight pore even at room temperature. These insights open up opportunities to engineer nanopore selectivity based on the molecular degrees of freedom.

[1] Department of Chemistry and Biochemistry, University of Maryland, College Park, MD 20742, USA. [2] Department of Mechanical Science and Engineering, University of Illinois at Urbana-Champaign, Urbana, IL 61801, USA. [3] Materials Science and Engineering Division, National Institute of Standards and Technology, Gaithersburg, MD 20899, USA. [4] Maryland NanoCenter, University of Maryland, College Park, MD 20742, USA. ✉email: yhw@umd.edu

Nanopores play an important role in chemical separations and selective mass transport that underlie many basic biological functions and industrial processes[1–3]. Biological systems, for example, have evolved a diverse array of specialized nanoscale protein channels that can allow only selected ions and molecules to cross the cell membrane[4]. Synthetic porous materials, such as zeolites, have also demonstrated an impressive level of molecular sieving capabilities for chemicals, allowing the separation of minor components from heterogeneous mixtures[5]. It is generally believed that the selectivity enabling such separations arises from the ability of molecules with smaller kinetic diameter (KD) to enter the pore while larger ones are excluded[6–9].

However, in examining nanopores defined by single-wall carbon nanotubes (SWCNTs), we find unambiguous evidence that molecules can adapt their conformation to enter smaller pores. The SWCNT nanopores are chemically inert, structurally rigid (Young's modulus >1 TPa)[10], and atomically smooth cylinders, featuring inflexible pores that are well-defined by the nanotube cylinder that is constructed from a conjugated sp[2] carbon lattice. These nanopores are tunable in size within the sub-nm range based on their individual atomic structure (i.e., nanotube chirality, as defined by a pair of integers $(n,m)$)[11], and they exhibit intriguing molecular transport properties[12,13]. Many of these SWCNT nanopores have a pore size comparable to small molecules. For example, (6,5)-SWCNT has a van der Waals pore size of just 0.422 nm, which is even smaller than n-hexane (KD ≈ 0.43 nm)[14,15] and cyclohexane (KD ≈ 0.60 nm)[15,16]. Additionally, the excitonic photoluminescence (PL) of semiconducting SWCNTs is sensitive to both the exterior and interior environments of the hollow nanotube[17,18] due to changes in the dielectric microenvironment[19], as well as from molecule-induced strain[20].

Here, we show that n-hexane is able to enter (6,5)-SWCNT, while cyclohexane is excluded. This is despite the fact that plausible filling configurations of both molecules suggest that neither molecule should be able to enter the rigid pore when observed from the view of KD (Fig. 1a, b). Our series of experiments further confirm that even a trace amount of 0.1% n-hexane in 99.9% (by volume) cyclohexane can be selectively captured and removed from the mixture. In contrast, for SWCNT pores only 0.025 nm larger, both molecules are able to enter and the selectivity is lost. By capturing the optical response of the nanotube, as schematically illustrated in Fig. 1c, and combining ab initio molecular dynamics simulations, we uncover a molecular level of insights for the nanopore selectivity.

## Results

To prepare samples of molecule-filled SWCNTs, we first thermally oxidized the nanotubes to open their ends, and then incubated these end-opened SWCNTs in cyclohexane or n-hexane (see "Methods" for details). This opening step is required as the ends of raw SWCNTs are typically capped or blocked, which would prevent the molecules from entering the pore[18,21]. We note that the nanotubes retain their structural integrity during this oxidative opening process. The amount of oxidative defects introduced, if any, is negligible, as evidenced by the nearly unchanged Raman D/G peak ratio (Supplementary Fig. 1a) and bright PL from the end-opened nanotubes (vide infra). After exposure to either n-hexane or cyclohexane, the opened nanotubes were then stabilized as individual particles in water by the surfactant sodium deoxycholate for ensemble measurements or deposited on a substrate for single nanotube hyperspectral imaging. As controls, we prepared both empty (end-capped) SWCNTs and water-filled end-opened nanotubes. The water-filled SWCNTs are a necessary control because the particle dispersion process required to achieve individualized nanotubes is also known to cut the nanotubes shorter and cause them to fill with water[17,21]. To prepare both controls, we dispersed end-capped SWCNTs and separated those that remained intact from the water-filled end-opened structures based on their difference in buoyant density[17,21].

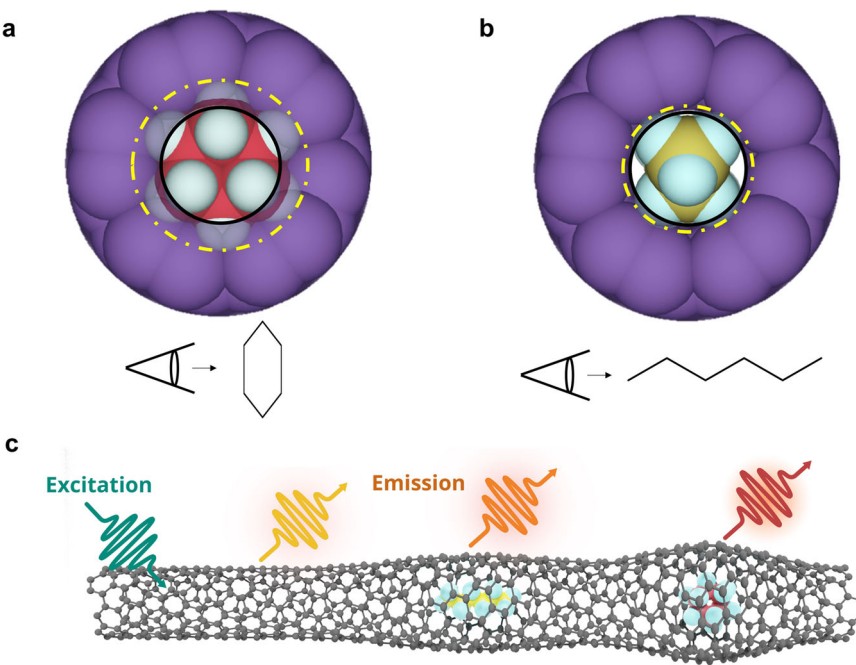

**Fig. 1 Photoluminescence response of a carbon nanotube to encapsulated molecules.** Cross-sectional views of **a** cyclohexane and **b** n-hexane with respect to a (6,5)-SWCNT (purple cylinder). The nanotube pore (as indicated by the black circle) is smaller than the molecule (as indicated by the dashed yellow circle). **c** The nanotube fluoresces at different wavelengths in response to different encapsulated molecules. The yellow arrow represents the PL emission from unfilled nanotube segments, while the PL emission from the filled positions are shifted depending on the encapsulated molecules.

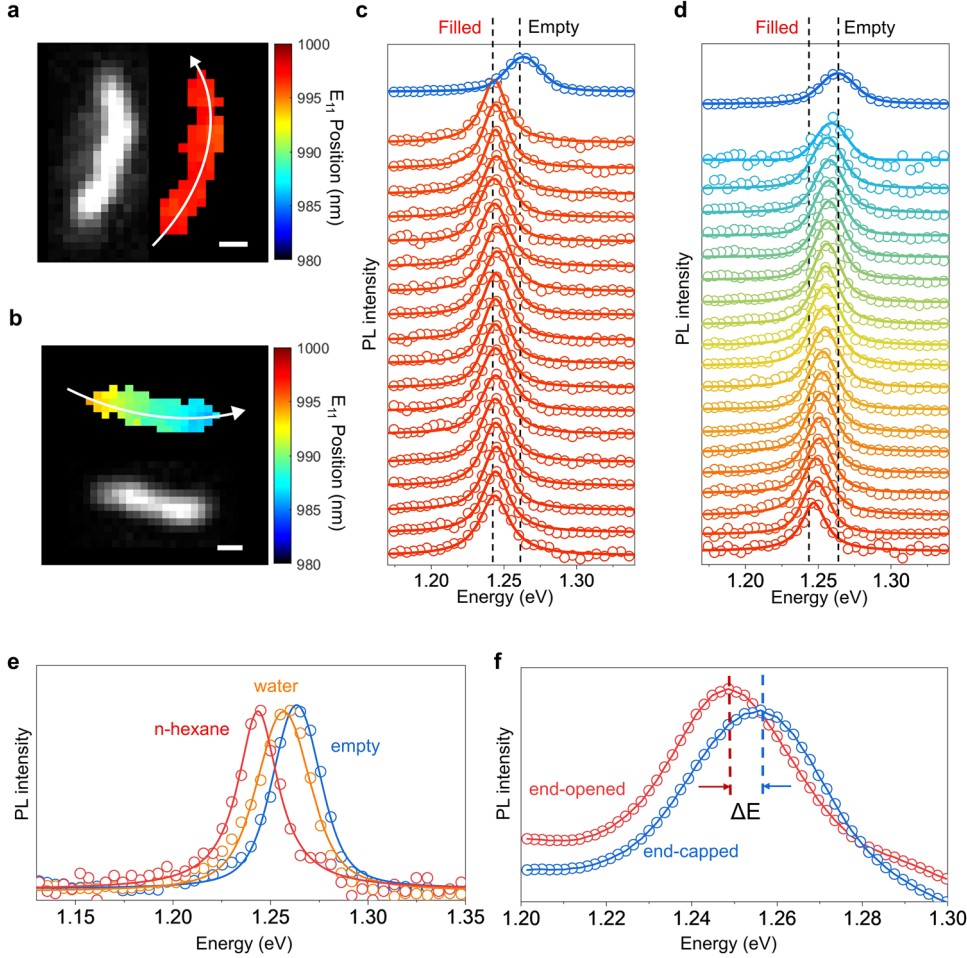

**Fig. 2 Hyperspectral imaging and photoluminescence spectroscopy of alkane-filled individual SWCNTs. a** Hyperspectral images of the $E_{11}$ PL from a single (6,5)-SWCNT that is filled with n-hexane along its length. Left: PL intensity image. Right: Map of the PL peak positions. **b** PL image of a (6,5)-SWCNT partially filled with n-hexane. Top: PL Peak position map. Bottom: PL intensity image. **c** PL spectra along the center of the n-hexane filled nanotube (indicated by the arrow in **a**) in comparison with an empty control. **d** PL spectra along the center of the nanotube partially filled with n-hexane (indicated by the arrow in **b**). **e** PL spectra from individual (6,5)-SWCNTs that are empty (blue), water-filled (orange), and n-hexane-filled (red). **f** Ensemble PL spectra of end-opened (red) and end-capped (6,5)-SWCNTs (blue) that had been incubated in n-hexane. Scale bars in **a**, **b** represent 500 nm. Note that in **a**, **b**, the peak position is from spectral fitting of each data set with a Gaussian function. The other spectra are fitted with a Voigt distribution function.

**Hyperspectral imaging of individual, molecule-filled SWCNTs.** We confirmed molecular filling by comparing the nanotube PL for the n-hexane exposed samples against both the empty and water-filled nanotube controls. Using hyperspectral imaging[22,23], we directly measured and mapped the PL response along the lengths of individual nanotubes. We then fit the spectrum captured on each detector pixel with a Gaussian function (see Supplementary Note 1 and Supplementary Fig. 2) and plot the peak PL emission wavelength as a pseudo-colored map along with the intensity image (Fig. 2a, b). Generally, hexane-filled (6,5)-SWCNTs emit at ≈998 nm, in contrast with the empty and water-filled (6,5)-SWCNTs, which emit at ≈982 nm and ≈987 nm, respectively (Fig. 2a, c, e). We find that hexane-filled (6,5)-SWCNTs are red-shifted by $(12 \pm 1)$ meV from the empty control and by $(8 \pm 2)$ meV compared with the water-filled counterpart. In some instances (21 out of 81 individual long nanotubes that were analyzed), we observed partially filled (6,5)-SWCNTs, as shown in Fig. 2b, d. The PL peaks shift from 994 nm to 987 nm at different points along the nanotube length, suggesting a single file of n-hexane molecules intermittently spaced along the nanotube. A simplified estimate of the numbers of n-hexane molecules in each pixel region is provided in Supplementary Note 1. More examples of empty and partially filled

(6,5), (8,3) and (7,5)-SWCNTs are shown in Supplementary Fig. 3.

**Origin of PL Shift.** We note there are two possibilities that could produce the observed shifts in the (6,5)-SWCNT spectra: surface adsorption or endohedral filling. If the PL shift were due to the surface adsorption of n-hexane on the nanotubes, then we would expect there to be no observable difference in the PL spectra of end-capped versus end-opened nanotubes. However, we find that the PL of the end-opened nanotubes is clearly redshifted compared to that from the end-capped nanotubes after both samples had been incubated with n-hexane (Fig. 2f). This result confirmed that the PL shift is caused by endohedral encapsulation of n-hexane molecules. Furthermore, spectroscopy studies confirm the fact that without opening the ends of the nanotubes, the interior channel is inaccessible for filling regardless of how large the pore is relative to the molecule (Supplementary Figs. 4, 5), which, again, supports that surface adsorption of hexane or cyclohexane is not the cause of the observed spectral shifts.

Molecular filling can induce strain on the nanotube[24] as well as change its interior dielectric environment[19], both of which may cause the observed PL shifts. However, strain-induced PL shifts

are strongly dependent on the nanotube mod ($n$-$m$, 3). Expansive radial strain is predicted to shift the emission of mod ($n$-$m$, 3) = 1 SWCNTs to the red (i.e., longer wavelength emission) and mod ($n$-$m$, 3) = 2 SWCNTs to the blue when compared to an unstrained nanotube[20,25]. In contrast, the dielectric environment is expected to induce the same effect on all nanotubes of the same diameter. We can differentiate these effects of strain vs. the dielectric environment by comparing n-hexane, cyclohexane, and water-filled nanotubes of different mod and relatively larger pore size. We note that water filling also causes a spectral shift due to the dielectric effect, but it features little to no strain since the molecular size of water is relatively small[17,21]. We observe that PL from (8,4)- and (7,6)-SWCNTs, both being mod 1, are redshifted when incubated with cyclohexane or n-hexane compared with water-filled controls (Supplementary Fig. 6). In contrast, mod 2 nanotubes, including (8,3), (7,5), and (9,4), show blueshifts. These trends closely follow the theoretical model by Yang et al.[20], which predicts a mod-dependent, radial strain-induced electronic effect. The larger the strain, the larger the change in the band gap energy ($\Delta E$), following

$$\Delta E \propto (-1)^{[\text{mod}(n-m,\,3)]+1} \cdot \sigma \cdot \sin(3\theta) \tag{1}$$

in which $\sigma$ is the radial strain, $\theta$ is the SWCNT chiral angle, and mod ($n$-$m$, 3) is the nanotube mod. In particular, the Fermi point is pushed away from or closer to the Brillouin zone vertices due to expansive strain depending on the mod of specific SWCNT species[20]. The mod-dependent behavior we observe (Supplementary Fig. 6) strongly suggests that strain induced from molecular encapsulation, rather than change in the local dielectric environment, is the primary cause of the observed spectral shifts.

We also conducted Raman spectroscopy to further confirm that the observed PL shift is indeed induced by strain (see Supplementary Note 3). We observed a $\approx 3\,\text{cm}^{-1}$ downshift of the G-band (Supplementary Fig. 1b) and $\approx 4\,\text{cm}^{-1}$ upshift of the radial breathing modes (RBMs) of the n-hexane-filled (6,5)-SWCNTs (Supplementary Fig. 1c) compared with the water-filled control. These shifts are indicative of n-hexane generating strain on the nanotube sidewalls, consistent with previous observations[24,26].

We defined the spectral shift between n-hexane- and cyclohexane-incubated SWCNTs as $\Delta E_{11} = E_{11,\text{cyclohexane}} - E_{11,\text{n-hexane}}$, and plot $\Delta E_{11}$ as a function of the pore diameter for each filling molecule (Fig. 3, van der Waals pore sizes are provided in Supplementary Table 1). We find that for mod 1 nanotubes ((6,5), (8,4), and (7,6)) $\Delta E_{11}$ becomes more negative with increasing pore diameter, as shown by the orange curve in Fig. 3, while for mod 2 chiralities ((8,3), (7,5), and (9,4)) $\Delta E_{11}$ becomes more positive. These trends unambiguously support a strain-induced electronic effect[20] due to the molecular filling. Interestingly, we observed a $\Delta E_{11}$ of $\approx 0$ meV for (6,5)-SWCNT, while for larger diameter nanotubes $\Delta E_{11} \neq 0$ meV. This occurs because when incubating (6,5)-SWCNT with cyclohexane it was not cyclohexane, but trace contaminant n-hexane that filled the nanotube. Even though the labeled purity for the commercially available high purity cyclohexane is 99.9%, n-hexane exists as a trace impurity that is difficult to remove completely, as we confirmed by composition analysis using gas chromatography coupled with a mass spectrometer (GC-MS) (Supplementary Fig. 7). In contrast, both n-hexane and cyclohexane can fit in larger diameter SWCNT hosts, such as (8,3), (7,5), and (8,4), and cause the mod-dependent PL shifts (Fig. 3).

## Molecular sieving of n-hexane from cyclohexane. As cyclohexane and n-hexane are nearly identical in chemical identity, the pair serves as an excellent model system to challenge the ability of nanotube pores for possible molecular sieving. We prepared a

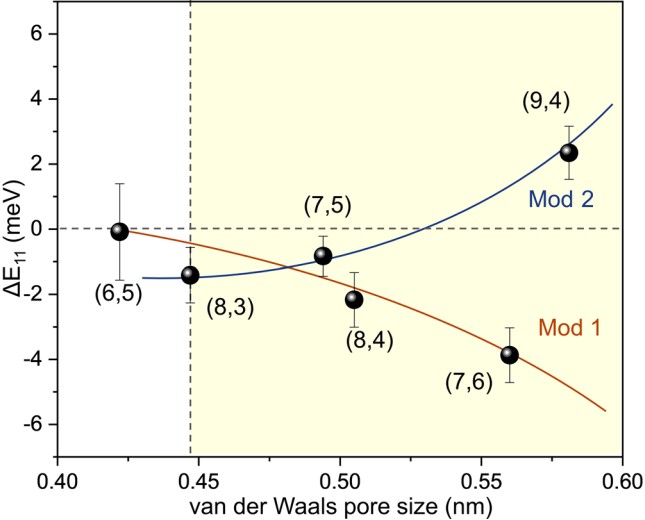

**Fig. 3 Molecular filling of nanotube pores is size dependent.** The PL energy differences between cyclohexane (99.9%)- and n-hexane-incubated end-opened SWCNTs ($\Delta E_{11} = E_{11,\text{cyclohexane}} - E_{11,\text{n-hexane}}$) obtained from experiments are plotted as a function of the van der Waals pore size of the nanotubes. Both n-hexane and cyclohexane can fit in those nanotube pores in the yellow colored area, while only n-hexane can enter the (6,5)-SWCNT pore, suggesting the existence of a threshold pore size below which n-hexane can enter but cyclohexane cannot. Note that the curves are added to guide the eye. The curves connect nanotubes of Mod 1, with mod($n-m$, 3) = 1, and Mod 2 (mod($n-m$, 3) = 2). The error bars represent the standard deviation of the $E_{11}$ emission peak position measured from multiple different SWCNT samples. Uncertainty in the calculated points are represented by error bars equal to one standard deviation.

mixture of 0.1% (by volume) n-hexane in 99.9% cyclohexane solution, in which we incubated end-opened SWCNTs composed of primarily (6,5)-SWCNTs at room temperature (Fig. 4). After incubation with different amounts of SWCNTs, the nanotubes were removed from the solvent mixture by filtration and the purity of cyclohexane was quantified by GC-MS. Figure 4b shows the cyclohexane purity (%, vol/vol) and n-hexane concentration (μmol/mL) as a function of the SWCNT mass added per unit volume. Note that the removal of n-hexane occurs only with open-ended SWCNTs (Supplementary Figure 8a). These experiments demonstrate that the n-hexane was selectively removed by the addition of the end-opened SWCNTs. We estimated the percentage of the pore volume occupied by n-hexane inside the (6,5) nanotubes was in the range of (52–82)% for the SWCNT concentrations measured, as shown in Fig. 4. The detailed calculation is described in Supplementary Note 2.

Our series of experiments provide unambiguous evidence that (6,5)-SWCNTs can be selectively filled with n-hexane. However, the widely used concept of KD failed to capture this phenomenon. This observation thus challenges our current understanding of the nanotube pore and may have broad implications for nanopore applications where molecular filling is the critical first step to selectivity and transport.

## Molecular dynamics and ab-initio molecular dynamics simulations. To understand how hexane enters the tight pore, we performed molecular dynamics (MD) and ab initio MD simulations. We terminated the nanotube ends with hydroxyl (-OH) groups, as expected from oxidative end-uncapping, and held the tube in place with graphene walls. We then kept the SWCNT in an n-hexane/cyclohexane bath at 300 K, as shown in Supplementary Fig. 9, and modeled the system as an isothermal-isobaric

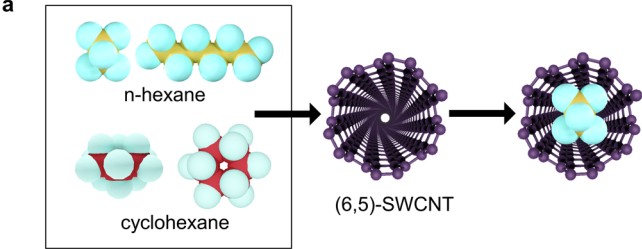

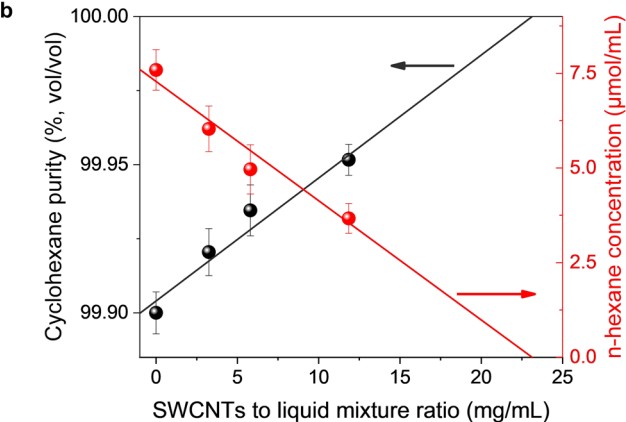

**Fig. 4 Molecular sieving of n-hexane from cyclohexane. a** Schematic of the molecular sieving of n-hexane and cyclohexane by (6,5)-SWCNTs. Only n-hexane molecules can enter the nanopore of (6,5)-SWCNTs. Note that for clarity, the carbon atoms are displayed in two different colors: golden (n-hexane) and red (cyclohexane). **b** Molecular sieving of n-hexane from 99.9% (volume/volume) cyclohexane by (6,5)-SWCNTs. Plotted are cyclohexane purity and n-hexane concentration as a function of the nanotube mass per unit volume added to purify the mixture. The cyclohexane purity was determined by quantitative GC-MS. The two trendlines are added to guide the eye assuming linear extrapolation. Uncertainty in the calculated points are represented by error bars equal to one standard deviation.

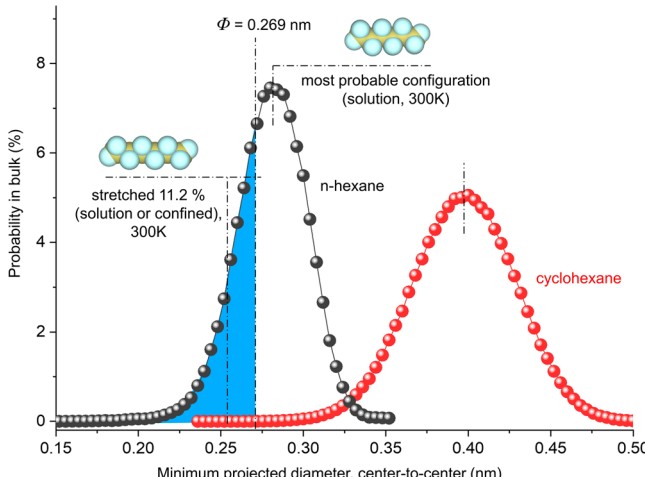

**Fig. 5 Molecular dynamics simulations revealing that n-hexane adapts a stretched molecular configuration to enter a tight pore.** Plotted are the distributions of the minimum projected diameter for each stereoisomer of n-hexane (black dots and line) and cyclohexane (red dots and line) in bulk solution at 300 K, which are sampled from a total of 222039 and 1131188 molecular configurations for n-hexane and cyclohexane, respectively. The molecular models are also shown for the most probable configuration of n-hexane in solution and in its stretched state (in bulk solution or under confinement of the (6,5)-SWCNT pore). The n-hexane molecule extensionally stretched 11.2 % with a minimum projected diameter 0.251 nm which is smaller than the accessible pore size of the (6,5)-SWCNT ($\Phi = 0.269$ nm). The overall probability that any particular free n-hexane molecule exists in a conformation sufficiently narrow to enter the (6,5)-SWCNT interior, calculated through the ratio of the integral of the shaded blue area in the figure to the entire minimum projection curve, is 28.5%, as highlighted by the blue shaded area. In contrast, there is zero probability for cyclohexane to exist at a molecular configuration smaller than the pore size of (6,5)-SWCNTs.

ensemble, in which the number of molecules, pressure, and temperature were held constant. Because the curvature effects[27] become so pronounced in these small diameter nanotubes, the concept of van der Waals radii of atoms[28,29], which model atoms as hard spheres, and the kinetic diameter of molecules[15], which are calculated based on the van der Waals radii, cannot fully capture the nanopore-molecule interactions. We incorporate these effects by performing ab initio MD where the electronic structure calculations are combined with all atom MD (see "Methods" for details) and calculate the projected radial distance ($\Delta$) between n-hexane atoms and the SWCNT carbons in the molecule filled nanotube (Supplementary Figure 10). By subtracting $\Delta$ from the nanotube diameter we can define an accessible pore size ($\Phi$) that can be directly compared with the minimum projected diameter of the filling molecule (Supplementary Figure 11). For the n-hexane filled (6,5)-SWCNT, we obtain a $\Delta$ of 0.244 nm and an accessible pore size of 0.269 nm for (6,5)-SWCNT. Interestingly, n-hexane enters (6,5)-SWCNT only when it is in the stretched state, whereas the un-stretched n-hexane is excluded from the pore of (6,5)-SWCNT (Supplementary Fig. 12, Supplementary Movies 1, 2). Compared to the relaxed state (i.e., the most probable configuration) in the bulk solution, the n-hexane molecule is stretched by 11.2% (elongated along the length) inside the (6,5)-SWCNT pore (Supplementary Table 2). MD simulations on (6,5)-SWCNT, (8,3)-SWCNT as

well as other SWCNTs (Supplementary Note 5 and Supplementary Movies 3–6) corroborate experimental observations (shown in Fig. 3) that there exists a threshold pore diameter below which only n-hexane enters the pore whereas cyclohexane cannot.

To further understand the stretched molecule size, we calculate the minimum projected diameter of n-hexane in each of its different molecular configurations, or stereoisomers, and observe that the most probable configuration occurs at a minimum projected diameter of 0.28 nm (Fig. 5). Importantly, we find that the stretched configuration of n-hexane (with a minimum projected diameter of 0.251 nm) exists in the bulk solution at 300 K and overall there is a 28.5% probability (blue-shaded area in Fig. 5) for a free n-hexane molecule to exist in molecular configurations that are smaller than the (6,5)-SWCNT's accessible pore diameter (0.269 nm). In contrast, for cyclohexane, the minimum projected diameter for the most probable configuration is 0.398 nm and there is zero probability for the molecule to exist at a configuration that is smaller than the (6,5)-SWCNT's accessible pore size, which explains its complete exclusion from the nanopore.

## Discussion

KD is a concept that has been widely used to represent the molecular size in the field of molecular sieving[30–32]. Our work, however, provides an example that this static view of molecular size fails to capture the flexible nature of molecules in entering a tight space. The structural rigidity of SWCNTs allows us to study the foundation of such observation by isolating the flexibility of the absorbate (n-hexane or cyclohexane in our case) from the

pore deformation which could occur with flexible pores such as metal organic frameworks due to the structural motions of ligands and lower Young's modulus[33]. Instead of using the KD, we simulate the dynamic changes of the molecules in solution to obtain the minimal projected diameter, which resolves the discrepancy and better captures the molecular origin of such a molecular sieving phenomenon.

In summary, we experimentally observed molecular sieving between two competing molecules that are both larger than the pore size of a precision nanopore defined by a carbon nanotube. MD simulations show that n-hexane enters the tight nanopore at its stretched state (elongating by nearly 11.2%), which exists with an overall probability of 28.5% in the bulk solution at 300 K. The encapsulated molecules cause a strain-induced PL shift that we used as a sensitive optical signature to directly resolve the molecular filling along the length of individual nanotubes by hyperspectral imaging. These unexpected observations provide insights on nanopore selectivity[3] and suggest a strategy for molecular separation[1,2] by exploiting the molecular degrees of freedom.

## Methods

**Process to Uncap the Carbon Nanotubes**. To enable filling of SWCNTs, we modified a published procedure[24] to maintain the nanotube length while removing the capped ends. 60 mg of cobalt-molybdenum catalyst (CoMoCat) SG65 SWCNTs (Sigma Aldrich, lot MKBS9734V) (certain equipment, instruments or materials are identified in this paper in order to adequately specify the experimental details. Such identification does not imply recommendation by the National Institute of Standards and Technology (NIST) nor does it imply the materials are necessarily the best available for the purpose) were placed on a glass slide and oxidized at ≈300 °C in air for 30 min and kept at the same temperature under 0.1 MPa (1 atm) of argon for 30 min, resulting in a 15% mass loss.

**Endohedral filling of alkanes**. A powder of the uncapped SWCNTs (≈10 mg) was placed in a 10 mL round bottom flask with 5 mL n-hexane (≥99.0% Uvasol®, Sigma Aldrich, lot K49581572804) or 5 mL cyclohexane (99.9%, Fisher Scientific, lot 176413), which was then incubated at 55 °C for 48 h. The mixture was then incubated for 48 h at 55 °C, which is below the boiling points of both n-hexane and cyclohexane, with the aid of a condenser to minimize evaporation. Post incubation, the mixtures were filtered (Millipore VVLP membrane, 0.1 μm pore size) to separate the bulk liquid from the solid nanotubes. The resulting cake, (2–10) mg, was manually crumbled loose and placed in a vacuum chamber for 24 h at room temperature to evaporate any residual alkane sticking to the outer walls of the nanotubes.

**Carbon nanotube dispersion process**. For ensemble spectroscopy measurements, we dispersed ≈(1–2) mg nanotubes in 2 mL of 10 g/L sodium deoxycholate (DOC) in D$_2$O by ultrasonication with a power of 4 W and at 10 °C for 30 min, followed by centrifugation at 25,000 × $g$ (1717 rad/s, 16400 rpm) for 1 h (Eppendorf centrifuge 5417R). In the case of separating empty nanotubes from water-filled ones by density gradient ultracentrifugation (as described in the next section), the ultrasonication power was ≈0.9 W/mL solution.

To obtain long and molecule-filled nanotubes for hyperspectral imaging, we performed a superacid surfactant exchange (S2E), as previously reported[34], in order to avoid ultrasonication that can cut nanotubes. Briefly, the alkane-incubated SWCNTs were dissolved in chlorosulfonic acid (Sigma-Aldrich, 99.9%) at a concentration of ≈0.2 mg/mL and the mixture was then added drop-by-drop to a solution of 0.75 mol/L NaOH and 0.8 g/L DOC (Sigma-Aldrich, ≥97%) aqueous solution until the solution pH decreased to ≈8. To concentrate the SWCNTs, the pH of the solution was then tuned to ≈6 by adding HCl (1 mol/L) to coagulate the SWCNTs. The precipitates were then collected and redispersed in Nanopure™ water by increasing the pH to ≈7–8 with NaOH solution (1 mol/L). The final concentration of DOC was 20 g/L. Undissolved particulates were then removed by centrifuging the solution at 23,264 × $g$ (1467 rad/s, 14000 rpm) for 60 min on a benchtop centrifuge (Eppendorf Centrifuge 5810 R). (Safety note: the neutralization process of the chlorosulfonic acid is extremely aggressive and should be performed in a fume hood with proper personal protective equipment, including goggles, lab coats, and acid-resistant gloves, due to the generation of a significant amount of heat and fumes.)

**Separation of end-capped SWCNTs by density gradient ultracentrifugation**. To obtain the end-capped empty SWCNTs, we used a density gradient ultracentrifugation procedure[21] to separate the empty SWCNTs from those that were water-filled based on their density differences. Due to the absence of end-capped

empty SWCNTs in CoMoCat SG65, which was only available as chemically purified material, we used raw HiPco SWCNTs (NoPo Nanotechnologies, LOT: R118-0810) dispersed at a concentration 20 g/L (≈2 %) DOC/H$_2$O by mild sonication (45 min, 6 mm tip, ≈0.9 W/mL) then centrifugation for 2 h with a JA-20 rotor at 39,086 × $g$ (1885 rad/s, 18000 RPM) to prepare a stock solution for this purpose. The SWCNTs in 2% DOC were layered on top of 4 mL of 10% (mass/volume) iodixanol (OptiPrep, Aldrich), 1% DOC/H$_2$O and centrifuged using a VTi 65.2 rotor (Beckman-Coulter) operated at 414,829 × $g$ (6807 rad/s, 65000 RPM) for 1 h at 20 °C. The enriched empty SWCNTs were collected from the top of the band in the middle of the tube. The iodixanol was removed by ultrafiltration through a 30 kDa membrane (Millipore 8003).

**Characterization**. To probe the optical properties at the ensemble level, the filled-SWCNTs were further diluted in 2 mL of 10 g/L DOC in D$_2$O. Samples were diluted in 10 g/L DOC in D$_2$O to avoid the absorption of H$_2$O at ≈1200 nm, which is very close to the SWCNT PL emission range.

The PL was collected using a NanoLog spectrofluorometer (Horiba Jobin Yvon). The samples were excited with a 450 W Xenon source dispersed by a double-grating monochromator. The slit width set bandpass of the excitation and emission beams were both set to 10 nm. PL spectra were collected using a liquid-N$_2$ cooled linear InGaAs array detector. The emission spectra were collected with excitation light at the E$_{22}$ wavelength of each specific chirality. The integration time for the 1D and 2D spectra were 2 s to 60 s and 5 s, respectively. Note that all samples had an optical density at the E$_{11}$ band of less than 0.5 (A/cm), measured using a PerkinElmer Lambda 1050 spectrophotometer with a broadband InGaAs detector.

The Raman spectroscopy was performed using a LabRAM ARAMIS Raman microscope (Horiba Jobin Yvon) with an 1800 or 2400 groove/cm grating, 532 nm laser excitation (46 mW), and a 1.0 neutral density filter to prevent sample damage. The integration time was 1 s, taken 10 times in total. The dispersed nanotube solution was precipitated with ethanol and then deposited on a Si substrate, which simultaneously served as a reference with the Si peak at 520.7 cm$^{-1}$ during the measurement.

To demonstrate the selective filling, the samples were prepared for GC-MS analysis as follows. 50 μL of n-hexane was mixed with 50 mL cyclohexane to prepare a 0.1% n-hexane in cyclohexane (vol:vol) solution. A powder of end-opened SWCNTs (from CoMoCat SG65 nanotubes, which contains a mixture of chiralities, but the majority of the sample is (6,5)-SWCNT) with different masses was placed in 3.7 mL vials with magnetic stir bars. 2 mL of the 0.1% n-hexane in cyclohexane solution was added to each vial. The vials were sealed with a cap and Parafilm®. The whole system was stirred at room temperature for 7 days to ensure it reaches equilibrium. Note: the boiling points of n-hexane and cyclohexane are 68.7 °C and 80.7 °C, respectively. Therefore, a sealed system was used to minimize any potential bias possible from differential evaporation. After incubation with different amounts of SWCNTs, the nanotubes were removed by filtration using a Millipore VVLP membrane (0.1 μm pore size). The purity of cyclohexane was analyzed in a GC auto sampler vial within 5 min to minimize the evaporation.

The GC-MS measurements were performed on an Agilent 6890N system coupled with a JEOL high-resolution magnetic sector mass spectrometer (JMS-700 MStation) with an electron ionization (EI) source (70 eV). The mass spectrometer was operated in the mode of high scan speed with a mass range from (50 to 200) Daltons. A silica capillary column (Agilent DB-17MS, 30 m length, 250 μm I.D.) was used in the experiments with helium (at 1 mL/min) as the carrier gas. The analysis was performed as follows: injection volume = 1 μL, splitless mode for the 99.9% pure commercial cyclohexane (Fisher Scientific, certified ACS, Lot 176413) and split mode (ratio = 1:20) for the mixture of n-hexane (Sigma Aldrich, Lot K49581572) and cyclohexane (volume:volume = 0.1:99.9), the front inlet temperature was 250 °C, the column temperature was programmed from 40 °C at 2.0 min, increased to 150 °C at a rate of 25 °C/min, then increased to 280 °C at a rate of 50 °C/min, and held at 280 °C for another 1.0 min. We used the fragmentation patterns obtained by EI to identify the structures of ions observed in the mass spectra based on the NIST mass spectral library[35]. To obtain the absolute concentration of n-hexane after the SWCNTs were incubated in the liquid mixture, we used toluene as an internal standard with a volume concentration of 0.1% (volume/volume) to construct the calibration curve (Supplementary Fig. 8b). The concentration of n-hexane was then calculated with the following equation:

$$C_{\text{n-hexane}} = \frac{A_{\text{n-hexane}}}{A_{\text{toluene}}} \times \frac{C_{\text{toluene}}}{RF} \tag{2}$$

in which $C_{\text{n-hexane}}$ and $C_{\text{toluene}}$ are the concentration in units of μmol/mL for n-hexane and toluene, respectively, and $A_{\text{n-hexane}}$ and $A_{\text{toluene}}$ are the peak areas for n-hexane and toluene, respectively, and RF is the response factor.

**Hyperspectral photoluminescence imaging**. To collect single nanotube PL images, the filled-SWCNTs were dispersed in 10 g/L DOC in D$_2$O with an optical density (OD) of ≈0.1 at the E$_{11}$ peak wavelength of (6,5)-SWCNT, then diluted 10-fold with 10 g/L DOC in D$_2$O. 5 μL of the diluted solution was then drop-cast onto a polystyrene coated Au on Si substrate. The polystyrene layer acts as an insulating layer to prevent the SWCNTs from contacting with the Au, which would quench the PL. The Au layer is added as a mirror to double the excitation and emission

light. The hyperspectral imaging was performed on a custom-built microscope[22]. In the current experiments, we used an infrared optimized ×100 objective (LCPLN100XIR, numerical aperture (NA) = 0.85, Olympus) along with a continuous wave laser at 730 nm (Shanghai Dream Lasers Technology Co., Ltd.) or 561 nm (Jive$^{TM}$ Cobolt AB, Sweden) as the excitation light source. Fluorescent emission from the sample was filtered through a long pass dichroic mirror (875 nm edge, Semrock, USA), which removed the elastic laser scattering from the sample, and dispersed by a volume Bragg grating (VBG; Photon Etc, Inc. Montreal, Canada). Only the diffracted light with a narrow bandwidth of 3.7 nm was collected on the detector to form a spectral image.

We exported the PL spectra from the hyperspectral cube and read out the coordinate or location for each fluorescent nanotube. The $E_{11}$ peak position was obtained by fitting the spectrum from each pixel with a Gaussian profile (see Supplementary Note 1 and Supplementary Fig. 2) and we then reconstructed the peak positions along with spatial information as a color map image by MATLAB.

**Molecular dynamics (MD)**. MD simulations were performed to simulate the molecular filling and calculate the distribution of the minimum projected diameter (Fig. 5) of n-hexane and cyclohexane in bulk. These simulations used the Large-scale Atomic/Molecular Massively Parallel Simulator (LAMMPS)[36] MD toolkit. To simulate organic molecules in MD, we use the all-atom optimized potentials for liquid simulation (OPLS-AA) potential[37] to model carbon-carbon and carbon-hydrogen interactions. The atoms in the SWCNT are simulated using the Adaptive intermolecular reactive empirical bond order potential (AIREBO)[38]. Interactions between carbon in the SWCNT and carbon and hydrogen in hexane and cyclohexane are modeled using the Lennard-Jones potential[39] with parameters given in Supplementary Table 3. The parameters for carbon were taken from ref. [38] and hydrogen were taken from ref. [40] and oxygen were taken from ref. [41]. The SWCNTs are terminated with −OH groups, and the pressure is maintained at 0.1 MPa (1 atm) and temperature at 300 K to model the experimental conditions. To identify the threshold at which the hexane enters whereas cyclohexane cannot, MD simulations were performed for the cyclohexane, hexane, and SWCNT systems ((6,5)-, (8,3)-, (7,5)-, (8,4)-, (7,6)-, (9,4)-SWCNT) with the angle parameters for n-hexane and cyclohexane completely relaxed to account for large conformational changes in these molecules (Supplementary Figs. 9, 13). The configurations to calculate the distribution of the projected diameter in Fig. 5 were collected from the simulations of n-hexane/cyclohexane placed in a bath of volume $2.5 \times 2.5 \times 2.5$ nm$^3$ at 300 K and 0.1 MPa (1 atm). The simulations are equilibrated for 5 ns, before post-processing is done to bin the molecules based on their projected diameter. A time step of 0.25 fs is used, with data being sampled every 100 timesteps and the system was run until enough statistics were obtained to ensure a smooth distribution (n-hexane system was run for ≈125 ps and the cyclohexane system for ≈600 ps). The damping parameters used in LAMMPS for the thermostat and barostat are 2.5 fs and 25 fs, respectively, under the NPT ensemble during the production run.

**Density functional theory (DFT)**. DFT was used to optimize the geometries used in AIMD simulations (described in the next section) and to verify that the minimum projected diameter of n-hexane and cyclohexane in their relaxed state (most probable configuration) match with the results obtained from MD. These simulations are performed using the Vienna Ab initio Simulation Package (VASP)[42] with the interactions between nuclei and electrons defined by projector augmented-wave (PAW) method[43]. The Perdew−Burke-Ernzerhof (PBE) functional[44] is used for exchange correlation energy. A $1 \times 1 \times 4$ grid has been used in these simulations along with an energy cut-off of 400 eV. Convergence with respect to the grid size and the energy cut-off is shown in Supplementary Fig. 14. We note that using 400 eV, the energy difference is 2.91 eV from the converged value, which occurs at an energy cut-off of 500 eV, but the latter simulation is ≈40% more expensive, which is why 400 eV was chosen as a compromise. It was verified that the optimization of the n-hexane and cyclohexane geometries leads to positive vibrational frequencies only and that the projected diameters of the most probable configurations of n-hexane and cyclohexane are in agreement with those obtained from MD (0.28 nm and 0.4 nm, respectively).

**Ab initio molecular dynamic simulations (AIMD)**. AIMD simulations were performed to show that n-hexane moves into (6,5)-SWCNT only when it is stretched (Supplementary Fig. 12) and to calculate the accessible pore size of (6,5)-SWCNT (Supplementary Fig. 10). These simulations were performed for the n-hexane and (6,5)-SWCNT systems starting from two configurations: (1) n-hexane (both stretched and unstretched) is placed at the pore mouth; (2) stretched n-hexane placed inside the SWCNT, away from the hydroxyl groups at the pore mouth. The modeled (6,5)-SWCNT has 1-unit cell consisting of 364 carbon atoms, with an additional 44 atoms of hydroxyl groups (−OH) at the end of the nanotube. In total, the systems have 428 atoms with the volume of the simulation box being $22.5 \times 22.5 \times 80$ Å$^3$. The data files (Supplementary Data 1–4) for the coordinates of the systems used are supplied in the Supplementary File. The starting geometry of the stretched n-hexane molecule is calculated from MD simulations by relaxing all the angle parameters, allowing the molecule to freely stretch within the nanotube

pore. AIMD simulations were performed in VASP and all the parameters used were same as those used for geometry optimization in DFT (mentioned in the previous section). In addition, the isothermal-isobaric ensemble was used, with the temperature and pressure maintained at 300 K and 0.1 MPa (1 atm). For the system described in the first configuration, the simulations were performed until it was sufficient to determine whether the molecule was being excluded or not, which is ≈2 ps as shown in Supplementary Fig. 12.

## Data availability
The data that support the findings of this study are available from the corresponding author on reasonable request.

## Code availability
The custom MATLAB codes developed in the current study are available from the corresponding author on reasonable request.

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

## Acknowledgements

This work is supported as a part of the Center for Enhanced Nanofluidic Transport (CENT), an Energy Frontier Research Center funded by the U.S. Department of Energy, Office of Science, Basic Energy Sciences under Award No. DE-SC0019112, and through internal National Institute of Standards and Technology (NIST) funds. We also acknowledge the use of the parallel computing resources: (1) Blue Waters, which was provided by the University of Illinois and National Center for Supercomputing Applications supported by NSF awards OCI-0725070, ACI-1238993 and the state of Illinois, and (2) Comet at San Diego Supercomputer Center, which was provided by XSEDE under TG-CDA100010 allocation. The authors thank Y. Li for assistance with the GC-MS measurements. We also thank M. S. Strano, A. Taqieddin, A. Brozena, S. Faucher, and A. Pham for valuable discussions.

## Author contributions

Y.H.W. and N.R.A. conceived and directed the research. H.Q., X.W., P.W., and Y.L. performed the experiments, and J.F. sorted end-capped nanotubes. A.R. performed MD and AIMD simulations. All authors contributed to data analysis. H.Q. and Y.H.W. wrote the manuscript with inputs from all co-authors.

## Competing interests

The authors declare no competing interests.
