## [Peer Review File · Nature Communications]

REVIEWER COMMENTS

Reviewer #1 (Remarks to the Author):

The authors report selective filling of n-hexane inside the pore of single-walled carbon nanotubes (SWCNTs), which allows the separation of n-hexane molecules from the mixture with cyclohexane. The solvent incorporation is evaluated by photoluminescence (PL) measurements of SWCNTs, in which spectral shifts are observed depending on filled molecules. Computational simulations indicate that stretching of the n-hexane structures occurs for the penetration of this molecule in the tight pore of SWCNTs. The phenomenon is quite unique and important for advanced application development using defined nanopores. On the other hand, for publication, additional experiments are still necessary to prove the suggested mechanism, which are described below.

1) On page 5, the authors discuss that strain of the tubes is assigned as a primary cause for the observed spectral shifts. When such strain-induced deformation of the tube structures occurs as shown in Figure 1, Raman spectral shifts could be observed particularly in G-band appearing around $\sim 1590 \text{ cm}^{-1}$. Thus, Raman scattering measurements are needed to investigate the suggested strain effect.

2) The uncapping treatment could increase defects on the tube structures not only at the edge parts. Thus, Raman scattering measurements for the analysis of D-band ($\sim 1300 \text{ cm}^{-1}$) and G/D ratios should be needed. The amount of the defects may change the response to the strain, which would be observed in difference of the filling-induced PL shifting manner.

3) From the result in Figure 4, as the authors discussed, one can see that the increasing in the SWCNT concentration resulted in adsorption of the larger amount of n-hexane from the cyclohexane solutions. However, it is not easy to consider how much percentage of the pore volume of the tubes is filled by n-hexane molecules. The estimation need to be conducted and be described in the manuscript.

4) In hyperspectral imaging results shown in Figures 2b, 2d and S1, more detailed explanation is necessary for gradually-shifting PL peaks by relating to spatial resolution in this observation. For example, how many n-hexane molecules filled in each pixel region can be expected for the middle range of the observed wavelengths such as at 900 nm?

5) Desorption of n-hexane molecules filled in SWCNTs also needs to be monitored by PL measurements or hyperspectral imaging experiments to elucidate the physical adsorption (filling) process.

6) To verify the stretching n-hexane molecules in the tubes which is observed in simulations, experimental evidence is necessary to recognize the unexpected filling behavior. Infrared absorption measurements of the n-hexane-filled SWCNTs would provide information by analyzing bending vibration modes of n-hexane (together with its stretching vibration modes), which can be compared with the simulation results shown in Table S2.

7) In the Methods section (page 9, last sentence), the procedure of “incubated at 55 oC for 48 h under reflux” is unclear because the boiling temperatures of n-hexane and cyclohexane are 68.7 oC and 80.7 oC, respectively, as described in the manuscript.

Reviewer #2 (Remarks to the Author):

This paper proposes an interesting phenomenon that a linear molecule relatively larger than the pore size of the carbon nanotube will alter its configuration to enter the nanotube whereas a molecule with non-linear structure will be rejected due to the low degree of freedom. The observed experimental finding has been supported with theoretical simulations. However the following information and points are needed to be addressed for this paper.

1. Although the paper deals with the exclusion of cyclohexane, the Kinetic diameter of cyclohexane – is never mentioned.
2. Projected diameter of n-hexane, stretched and unstretched, as mentioned in Table S2 - How do these values (2.8 and 2.5 Angstrom) compare with its kinetic diameter (4.2 Angstrom). Why is a projected diameter so low? This merits a discussion in main paper.
3. Figure 1 – relative sizes of the hexane, cyclohexane with respect to (6,5) nanotube. Nanotube diameter was 0.42 nm and the n-hexane kinetic diameter as mentioned in the paper was 0.43 nm. But the relative sizes as shown in the figure 1 seem way off with nanotube being shown as much smaller than the n-hexane. And coming to point 2 above, if the diameter is so low, it should be easily able to enter the nanotube.
4. In figure 1, the legend reads blue arrow represents emission – please correct this, as the bluish color in the figure is excitation.

5. According to the statement made in Page 5, second paragraph that “water filled which are expected to feature little to no strain since the molecular size of water is relatively small”, how would the authors explain the spectral shift of water-filled SWCNT from empty SWCNT shown in Figure 2e and Figure S4 including the larger diameter ones which are not supposed to show a change as the diameter of water is small? The authors need to amend the statements and explain clearly.

6. It needs to be explained why one nanotube mod can cause a redshift while the other one mod will cause a blueshift. In page 5, it reads “radial strain-induced electronic effect”. However since this is the whole basis of the measurement, it could be explained in more detail.

7. Figure 4 – Figure 4a gives same information as in figure 1a and b, but a bit more clearly. The repetition can be avoided but make the figure 1a and b bit more clear (comment 4 above).

8. The authors have introduced an accessible pore size which is used as a cut-off value for the probability evaluation. In the main context they are saying the value is about 0.269 nm, which is smaller than the size of cyclohexane with any configurations. In Page 8, it seems that the authors use the n-hexane filled SWCNT system to calculate the project radical diameter first, then derive the accessible pore size. If so, how can you use this value which is acquired from the n-hexane system to evaluate the transport of cyclohexane molecules? When you calculate the project radical diameter, the interaction of n-hexane with the SWCNT has been taken into consideration. It will be invalid when you examine a different molecule.

Figure 5 – it is probably hard to read the message of the figure here. Let us take cyclohexane – the probability at the peak of cyclohexane is close to 5%, whereas the legend reads, “there is zero probability for cyclohexane to exist at a molecular configuration smaller than the pore size of (6,5)-SWCNTs.”. Can this be explained?

9. Figure 2 – shows the optical images of the nanotubes, along with the spectral images. SEM and/or images of the nanotubes are needed to provide the information of the morphology of the nanotubes

10. Does the filling of the nanotube change the nanotube topography? AFM images before and after the filling could hold a key to this information.

Reviewer #3 (Remarks to the Author):

In this combined experimental/computational study, Qu and co-workers investigate the potential of several single-walled carbon nanotubes (SWCNTs) to separate n-hexane from cyclohexane based on the SWCNT's size selectivity. Using the photoluminescence shift of the first electronic transition upon adsorption, they conclusively demonstrate that n-hexane and cyclohexane can both be adsorbed inside all nanotubes except for the (6,5)-SWCNT, in which only n-hexane is adsorbed. This observation, which is deemed counterintuitive by the authors as both cyclohexane and n-hexane

have larger kinetic diameters (KDs) than the van der Waals (vdW) pore size of the (6,5)-SWCNT, is explained using both classical and ab initio molecular dynamics (MD) simulations. These simulations point towards the ability of n-hexane to stretch and hence reduce its effective diameter to fit into the SWCNT. As cyclohexane is observed not to undergo a sufficient stretching, it is completely excluded from the nanotube, resulting in the selective filling of n-hexane mentioned in the title.

While this study conclusively shows the ability of (6,5)-SWCNT to sieve n-hexane from cyclohexane, the authors' focus on the vdW pore size and the KDs of the molecules to yield the apparent contradictory adsorption of n-hexane in (6,5)-SWCNTs is a very static view of molecular sieving. In the last decade, this static view has been continuously challenged, as both the vdW pore sizes and the KDs of molecules are inherently dynamic parameters that change over time and are influenced by external conditions such as temperature, pressure, and the solvent in which the molecules and SWCNTs are present. This is true for all materials, especially if they allow for substantial flexibility (see for instance 10.1021/jacs.7b01688). This flexibility, in this case exhibited both by the adsorbed molecules and the strained SWCNTs, is also demonstrated here in Figure 5: while the projected diameter is a property that shows large fluctuations and is characterized by a broad distribution, the KD of each molecule represents only a single value that does not take into account this broad distribution. As a result, the comparison between the KDs and the vdW pore size in the introduction that leads to the apparent contradiction is rather artificial.

Based on this assessment, the novelty of the manuscript is the successful sieving of n-hexane from cyclohexane in (6,5)-SWCNTs, which is successfully demonstrated both on experimental and computational grounds. In contrast, the apparent incompatibility between the observed n-hexane adsorption and the vdW pore size of the SWCNT that is smaller than the KD of n-hexane, an incompatibility that is emphasized even in the abstract, is artificial. I would therefore urge the authors to solely focus on the successful sieving properties of the (6,5)-SWCNT and the explanation from their MD simulations as shown in Fig. 5. However, this also limits the novelty of this manuscript and I'm therefore unsure whether it would be a good fit for the broad audience of Nature Communications. Below, more detailed suggestions for improvement are given that the authors may wish to take into account upon revision.

1. Since the reported vdW pore sizes of the SWCNTs and the KDs of the adsorbed molecules play a crucial role in the current version of the manuscript, it would be instrumental to verify that the procedure leading to the vdW pore sizes of Table S1 is a sensible one. I doubt whether the geometry of graphite is representative for the interactions present in the strongly curved geometry of the SWCNTs and hence whether the vdW radii can be readily extracted using this procedure. This is especially relevant since the observed difference between the n-hexane KD (0.43 nm) and the (6,5)-SWCNT vdW pore size (0.422 nm) is minimal, possibly invalidating this claim when slightly altering the procedure to determine the vdW pore size. In this respect, it would also be instructive to report the KD of cyclohexane, as well as the procedure with which these KDs are determined. Finally, how are the vdW pore sizes of the SWCNTs influenced by the different experimental steps (thermal

oxidation, incubation in solution, filtration)? Given the observation of strain and a PL shift upon adsorption, the altered interactions between the SWCNT and the adsorbed molecules will most likely also change the vdW pore size.

2. The statement on page 6 that “cyclohexane and n-hexane are nearly identical in structure” seems misplaced or insufficiently precise, given the substantial difference in their geometry.

3. How long were the SWCNT present in the n-hexane and cyclohexane solutions before they were removed (see first paragraph of page 7)? Does this incubation time change the sieving properties?

4. At the end of the first paragraph of page 7, the authors suggest that “extrapolation of the experimental curve in Figure 4b suggests that 24.1 mg/mL of end-opened SWCNTs can completely strip the n-hexane from the mixture to attain 100.0% pure cyclohexane.” This linear extrapolation of Figure 4b from 99.95% to 100% purity is not warranted based on the available data, which seem to indicate a flattening of the purity upon higher SWCNT concentration. If retained, this claim of 100% purity should also be experimentally validated.

5. On page 5 and in Figure 5, the authors illustrate the dynamic character of the projected diameter of the two molecules, which seems to be a better descriptor of the system than their static KDs. However, some questions remain on this part. What is the “relaxed state in the bulk solution” the authors refer to, and how is it determined? How were the stereoisomers determined to bin the MD generated molecules into – solely based on their projected diameter? How did the authors come to the 222,039 and 1,131,188 molecular configurations for n-hexane and cyclohexane, respectively? Can these molecular configurations be considered independent, or do they stem from a single simulation for each molecule? If they stem from a single simulation, it is necessary to check whether the observed distribution can be reproduced when using different initial conditions during the MD simulation. It would also be beneficial if the authors could provide geometry files for different relevant molecular configurations of n-hexane. Finally, the authors refer to “the stretched configuration of n-hexane” and an associated 2.7% probability to encounter it. Given the continuous character of the stretching observed in Figure 5, I would suggest to focus only on the probability of finding n-hexane with configurations that are smaller than the (6,5)-SWCNT’s accessible pore diameter. What does the blue shaded area in Fig. 5 represent?

6. In the Methods section, the paragraph on “Molecular Dynamics (MD) and Ab initio Molecular Dynamics (AIMD)” is in its current form insufficient to reproduce the simulations and to validate whether the simulations are sufficiently accurate to back up the reported results. Below, I have listed seven such issues, but it may be expected more are present.

First, it is not always clear which parts of this section pertain to the force-field MD simulations and which to the AIMD simulations.

Second, for the AIMD simulations, a 1x1x4 k-point grid and an energy cut-off of 400 eV have been used. To verify whether these computational values are sufficient, it is important to confirm that the obtained results are converged with respect to the grid size and the energy cut-off, which is rather low. Especially the 1x1x4 k-point grid seems odd, as the models of Fig. S10 are defined by one long direction (along the nanotube axis) and to shorter ones. As the k-point grid is defined in the reciprocal space, I would therefore expect a k-point grid of the type 1x4x4, for instance.

Third, how many atoms and molecules were present in these systems, and how large were the systems? Ideally, input files would be present for all relevant systems. This is especially relevant to reproduce the AIMD simulations with the stretched and unstretched configurations of n-hexane, as these descriptors do not fully determine the molecular configuration.

Fourth, the molecules inside the nanotube were allowed to stretch, but only the angle parameters were relaxed. Why weren't the other internal parameters, such as bond lengths, relaxed, and why wasn't the nanotube allowed to relax?

Fifth, during the AIMD simulations, the molecules were placed at the pore mouth. How does the exact position and orientation of the molecules influence the adsorption or rejection of these molecules? In this paragraph, the authors should also refer to Fig. S10 instead of Fig. S9.

Sixth, how were the Lennard-Jones parameters of Table S3 determined – those are not discussed in Ref. 33? This is especially relevant since the Lennard-Jones parameters play a pivotal role during adsorption in tight pores – small changes in the sigma parameters can be expected to lead to substantial differences in the results of for instance Figure 5.

Finally, what were the thermostat and barostat relaxation times, and the total simulation times for all simulations performed in this manuscript?

7. A color bar is missing for Figure S4, panes (a) and (b).

Point-by-Point Response to Reviewers' Comments

(color codes: black – editorial and reviewers' original comments; blue – response; red – changes made to the revised manuscript)

Reviewer #1 (Remarks to the Author):

The authors report selective filling of n-hexane inside the pore of single-walled carbon nanotubes (SWCNTs), which allows the separation of n-hexane molecules from the mixture with cyclohexane. The solvent incorporation is evaluated by photoluminescence (PL) measurements of SWCNTs, in which spectral shifts are observed depending on filled molecules. Computational simulations indicate that stretching of the n-hexane structures occurs for the penetration of this molecule in the tight pore of SWCNTs. The phenomenon is quite unique and important for advanced application development using defined nanopores. On the other hand, for publication, additional experiments are still necessary to prove the suggested mechanism, which are described below.

1) On page 5, the authors discuss that strain of the tubes is assigned as a primary cause for the observed spectral shifts. When such strain-induced deformation of the tube structures occurs as shown in Figure 1, Raman spectral shifts could be observed particularly in G-band appearing around $\sim 1590\text{ cm}^{-1}$. Thus, Raman scattering measurements are needed to investigate the suggested strain effect.

Response: We thank the reviewer for the suggestion. Raman spectroscopy is indeed valuable for characterizing the encapsulation of molecules in nanotubes based on the shift in the G-band and radial breathing modes (RBMs), as has been previously studied. For example, Streit et al. (*J. Phys. Chem. C* **122**, 21, 11577–11585 (2018)) found that the G-band downshifts and the RBMs upshift when the nanotubes are filled with tetracosane compared with water-filled control. Lebedkin, et al. (*Phys. Rev. B* **73**, 094109 (2006)) demonstrate the upshift of RBM upon applying compressive strain on SWCNTs.

To explore this effect in our system, we performed Raman spectroscopy on (6,5)-enriched SWCNTs to investigate the behavior of the G-band and RBM modes. The results are included in the Supplementary Information, which appears as Supplementary Note 3 and Supplementary Figure 1 (shown as Figure R1 in this response), along with a discussion in the text. Briefly, we observed the G-peak of the n-hexane filled (6,5)-SWCNTs to downshift by 3 cm^{-1} compared with the water-filled unstrained control (Figure R1a). This is similar to Streit et al.'s observation from a different system, supporting that n-hexane filling indeed induces strain on the nanotube sidewalls. Meanwhile, the RBM peak upshifts by 4 cm^{-1} in the hexane-filled SWCNTs compared with the water-filled nanotubes (Figure R1b). The hardening of the RBM is a further evidence of the strain generated by the encapsulated n-hexane.

We have revised the main text to include this new data (along with relevant experimental details in the Methods on p. 14), as follows:

“We also conducted Raman spectroscopy to further confirm that the observed PL shift is indeed induced by strain (see Supplementary Note 3). We observed a $\approx 3\text{ cm}^{-1}$ downshift of

the G-band (Supplementary Figure 1b) and $\approx 4 \text{ cm}^{-1}$ upshift of the radial breathing modes (RBMs) of the n-hexane-filled (6,5)-SWCNTs (Supplementary Figure 1c) compared with the water-filled control. These shifts are indicative of n-hexane generating strain on the nanotube sidewalls, consistent with previous observations.^{29,30} (p. 7, lines 1-6)

Figure R1. Raman spectra of n-hexane-filled (6,5)-SWCNTs in comparison with H₂O-filled and end-capped controls. **a**, Raman spectra for end-opened SWCNTs (black curve) and end-capped SWCNTs (red curve). **b**, Raman spectra of (6,5)-enriched SWCNTs that are filled with n-hexane (black curve) and H₂O (red curve). **c**, RBMs of the n-hexane-filled and H₂O-filled (6,5)-SWCNTs. Note that all the spectra are normalized to the G or RBM peak intensity.

2) The uncapping treatment could increase defects on the tube structures not only at the edge parts. Thus, Raman scattering measurements for the analysis of D-band ($\sim 1300 \text{ cm}^{-1}$) and G/D ratios should be needed. The amount of the defects may change the response to the strain, which would be observed in difference of the filling-induced PL shifting manner.

Response: We have added Raman spectra for the end-capped and end-opened SWCNTs in Figure R1c of this response, which appears as Supplementary Figure 1a in the revised Supplementary Information, along with a discussion in Supplementary Note 3. We observed only a slight increase of the D/G ratio from 0.021 to 0.032 upon the uncapping treatment, thus the effect of defects on the strain response and PL shifts should be negligible. Furthermore, these end-opened nanotubes feature bright PL, which is unambiguous evidence that the defect density is extremely low, as defects are known to quench nanotube PL (*Nano Lett.* **11**, 1636-1640 (2011)). We have added a note in the revised manuscript to clarify this point:

“We note that the nanotubes retain their structural integrity during this oxidative opening process. The amount of oxidative defects introduced, if any, is negligible, as evidenced by the nearly unchanged Raman D/G peak ratio (Supplementary Figure 1a) and bright PL from the end-opened nanotubes (*vide infra*).” – (p.3 line 20-21, p.4 line 1-3)

3) From the result in Figure 4, as the authors discussed, one can see that the increasing in the SWCNT concentration resulted in adsorption of the larger amount of n-hexane from the cyclohexane solutions. However, it is not easy to consider how much percentage of the pore volume of the tubes is filled by n-hexane molecules. The estimation need to be conducted and be described in the manuscript.

Response: We can estimate the percentage of the pore volume occupied by n-hexane from the data presented in Figure 4. We have added the details of this estimation as Supplementary Note 2 in the Supplementary Information. For this estimate, we assume all the SWCNTs in the solution are (6,5)-SWCNTs, though we note the nanotube raw material is synthesized as a mixture of different chiralities, with (6,5)-SWCNTs being the major component. We also considered the SWCNT purity (~77.1 wt.% based on the manufacturer's product specs).

Adsorption of n-hexane should result in the n-hexane molecules packing single-file inside the (6,5)-SWCNTs due to the size limitation. Therefore, we can estimate the percentage of the pore volume filled with n-hexane based on the length of the molecule and that of the SWCNTs. Since only stretched n-hexane can enter (6,5)-SWCNTs, the molecular configuration that fits the nanotube has a length of 1.14 nm, which can be calculated by adding the atom center-to-center distance (0.90 nm) of stretched n-hexane between the two end hydrogens, as reported in Table S2, with the combined van der Waal radii of the two hydrogen atoms being 0.24 nm at each end of the n-hexane (2×0.12 nm, taken from *Inorg. Mater.* **37**, 871-885 (2001)). When one n-hexane fills inside a nanotube, the length of this segment is roughly equal to the length of the stretched n-hexane. We can estimate when 100% filled, these nanotubes can encapsulate at least 2.07 $\mu\text{mol/mL}$, 3.68 $\mu\text{mol/mL}$, 7.52 $\mu\text{mol/mL}$ of n-hexane inside 2.51 mg/mL, 4.47 mg/mL, and 9.13 mg/mL (6,5)-SWCNTs, respectively. The actual amount of n-hexane removed was experimentally determined to be 1.56 $\mu\text{mol/mL}$, 2.63 $\mu\text{mol/mL}$, and 3.92 $\mu\text{mol/mL}$, respectively, as shown in Figure 4.

From the ratios, we obtained the percentage of the filled space to range from 52% to 82 % for the experiments performed here. These data suggest that the molecules are loosely packed inside the nanotube pore.

We have included this calculation to Supplementary Note 2 in the Supplementary Information, and briefly discussed the results in the main text, as follows:

“We estimated the percentage of the pore volume occupied by n-hexane inside the (6,5) nanotubes was in the range of (52-82) %, for the SWCNT concentrations measured, as shown in Figure 4. The detailed calculation is described in Supplementary Note 2.” (p. 8, lines 15-17).

4) In hyperspectral imaging results shown in Figures 2b, 2d and S1, more detailed explanation is necessary for gradually-shifting PL peaks by relating to spatial resolution in this observation. For example, how many n-hexane molecules filled in each pixel region can be expected for the middle range of the observed wavelengths such as at 900 nm?

Response: We thank the reviewer for raising this intriguing question.

First, we want to clarify that a (6,5)-SWCNT does not emit at 900 nm no matter what the encapsulated molecule or surfactant used in our setup. The (6,5)-SWCNTs in our experiment emit from 982 nm to 998 nm depending on the molecule.

In terms of estimating the number of n-hexane molecules in each pixel region, we have to consider the diffraction limit. The pixel size (~160 nm) is smaller than the diffraction limit, which stands at ~580 nm in our case (calculated from $\frac{\lambda}{2NA}$, where λ is the wavelength of the 980 nm emission from (6,5)-SWCNT, and $NA=0.85$ is the numerical aperture of the objective used). For this work we used long nanotubes (> 3 μm) to achieve sufficient PL spatial

resolution. It is not possible to completely deconvolute the complexity formed by so many emitting nanotube segments when they are partially filled. However, we may estimate the number of n-hexane molecules in each pixel based on the assumption that strain and PL shift correlation is linear.

In the case of n-hexane filling, the major cause of the PL shift is the strain, which is described by Yang's model (*Phys. Rev. Lett.* **85**, 154-157 (2000)) and other literature reports (*J. Phys. Chem. C* **113**, 571-575 (2008)). The PL shift (ΔE) is observed to be directly proportional to the strain (*Nano Lett.* **8**, 826-31 (2008)). Due to the single-file packing, we can consider the system as one dimensional such that the strain is linearly correlated with the number of filled molecules. If this assumption holds, the strain and resulting PL shift is then directly proportional to the number of n-hexane molecules within the nanotube.

By considering a linear scaling of the PL peak shift for the empty (0 %) up to the fully filled (100 %) nanotubes, we can obtain the percentage (P) of the nanotube filled with n-hexane from the E_{11} peak wavelength by the following relation:

$$P = kE_{11} - b,$$

where E_{11} is the PL emission peak wavelength, k and b are two constants due to the linear relationship of strain and resulting PL. We then substitute ($E_{11} = 982$, $P = 0\%$) and ($E_{11} = 998$, $P = 100\%$) into the equation to obtain $k (=0.0625)$ and $b (=61.375)$ for this case.

When one n-hexane fills inside a nanotube segment, the length of this segment is roughly equal to the length of the stretched n-hexane (1.14 nm, based on the atom center-to-center length of n-hexane, 0.90 nm, plus the van der Waal radii of the two hydrogen atoms, 2×0.12 nm = 0.24 nm). Based on these, the numbers of n-hexane can be derived. For example, for pixels that have an E_{11} peak of 990 nm in Figure 2b, we can calculate the percent filling of the nanotube is 50% n-hexane by plugging in $E_{11} = 990$ nm. We can further derive that on average there are ≈ 0.43 hexane molecules per nanometer length of (6,5)-SWCNT.

We have roughly estimated the number of n-hexane molecules within each pixel based on the PL shift of the nanotubes and have added this estimate in Supplementary Note 1. We refer to this addition to the manuscript on page 5 (lines 4) of the main text.

5) Desorption of n-hexane molecules filled in SWCNTs also needs to be monitored by PL measurements or hyperspectral imaging experiments to elucidate the physical adsorption (filling) process.

Response: We tried to demonstrate the desorption of the n-hexane molecules from the (6,5)-SWCNTs by monitoring the PL spectra 1 and 8 months after preparation of the sample (i.e., dispersing the nanotubes in solution using surfactant, then keeping samples in a drawer for 1 or 8 months) and comparing with freshly prepared sample, as shown in Figure R2a. We observed the PL peaks of these 2 aged samples are very similar to the freshly prepared sample in terms of their peak wavelength, suggesting the encapsulated molecules are stable inside the solution-dispersed SWCNTs. Additionally, as we hypothesized that the surfactant may block the pore to prevent n-hexane desorbing, we continuously monitored the PL of n-hexane filled (6,5)-SWCNTs that were placed in a vacuum for different amounts of time prior to dispersion (i.e., no surfactant present; Figure R2b). We then dispersed these samples in solution and measured the PL. We observed that the peak positions were the same and

distinct from the empty or water-filled nanotube controls. This suggests the stability of these encapsulated molecules inside the nanotubes. This may be due to the modification of the freezing and boiling point as a result of spatial confinement (*Nat. Nanotechnol.* **12**, 267-273 (2017); *J. Phys. Chem. Lett.* **5**, 408-12 (2014)). We conclude that desorption of the n-hexane molecules cannot be achieved on our current experimental setup, which require significant further advances in future studies.

Figure R2. PL spectra of n-hexane filled (6,5)-SWCNTs over time. **a**, PL spectra from a freshly prepared (blue) n-hexane filled (6,5)-SWCNT sample, and the same sample 1 month (black) and 8 months (red) after preparation. **b**, PL spectra from n-hexane filled (6,5)-SWCNTs that were placed in vacuum for different amounts of time prior to dispersion in solution.

6) To verify the stretching n-hexane molecules in the tubes which is observed in simulations, experimental evidence is necessary to recognize the unexpected filling behavior. Infrared absorption measurements of the n-hexane-filled SWCNTs would provide information by analyzing bending vibration modes of n-hexane (together with its stretching vibration modes), which can be compared with the simulation results shown in Table S2.

Response: We thank the reviewer for this intriguing suggestion. We indeed considered the possibility of analyzing the stretching of the molecules. Although we did not perform infrared absorption measurements, the Raman spectroscopy we performed on these systems do show that some stretching and vibrational modes of the confined molecules are different from measurements of bulk n-hexane (Figure R3). Unlike the multiple peaks of bulk n-hexane, the confined n-hexane-filled SWCNTs only shows one peak at 2892 cm^{-1} , which may correspond to a C-H stretching mode of the n-hexane, suggesting all other stretching may be limited due to confinement or the signal is too weak to detect due to the small amount of n-hexane trapped inside the SWCNTs.

In this work we employ single nanotube PL imaging measurements that allow us to selectively probe individual (6,5)-SWCNTs, which are typically found in a mixture of different chiralities. However, bulk (ensemble) measurements, such as Raman and infrared spectroscopies, will also include other chiralities beyond (6,5) nanotubes. Thus, the vibrational modes of n-hexane may depend on the specific confinement effects of different nanotube structures, which limits the useful conclusions we can make from this experiment.

Figure R3. Raman spectra. Blue: n-hexane in bulk, Red: n-hexane filled inside (6,5)-SWCNTs. Black: unfilled (6,5)-SWCNTs.

7) In the Methods section (page 9, last sentence), the procedure of “incubated at 55 oC for 48 h under reflux” is unclear because the boiling temperatures of n-hexane and cyclohexane are 68.7 oC and 80.7 oC, respectively, as described in the manuscript.

Response: We thank the reviewer for the note. Due to the high volatility of these two molecules at elevated temperature, we wanted to minimize the loss of liquid during this 48 h incubation. To minimize evaporation of the liquid, we used a condenser to keep the solution below the boiling points of both molecules. It is inaccurate and confusing that we wrote that the sample was prepared “under reflux.” We have corrected this error and added a note in the revised manuscript, as follows:

“The mixture was then incubated for 48 h at 55 °C, which is below the boiling points of both n-hexane and cyclohexane, with the aid of a condenser to minimize evaporation.” (p.12, line 5-7).

Reviewer #2 (Remarks to the Author):

This paper proposes an interesting phenomenon that a linear molecule relatively larger than the pore size of the carbon nanotube will alter its configuration to enter the nanotube whereas a molecule with non-linear structure will be rejected due to the low degree of freedom. The observed experimental finding has been supported with theoretical simulations. However the following information and points are needed to be addressed for this paper.

1. Although the paper deals with the exclusion of cyclohexane, the Kinetic diameter of cyclohexane – is never mentioned.

Response: We thank the reviewer for pointing out this omission. The kinetic diameter of cyclohexane has been reported to be 0.60 nm by multiple publications (*Phys. Chem. Chem. Phys.* **15**, 8795-8804 (2013); *J. Phys. Chem.* **97**, 1451-1454 (1993); *Zeolite molecular sieves: structure, chemistry, and use.* (Wiley, 1973)). We have included the KD and references in the revised manuscript:

“Our series of experiments demonstrate that n-hexane (KD ~0.43 nm)^{13,14} is able to enter (6,5)-SWCNTs with a van der Waals pore size of just 0.422 nm, while cyclohexane (KD ~0.60 nm)^{14,15} is excluded.” (p. 3, lines 1-3)

2. Projected diameter of n-hexane, stretched and unstretched, as mentioned in Table S2 - How do these values (2.8 and 2.5 Angstrom) compare with its kinetic diameter (4.2 Angstrom). Why is a projected diameter so low? This merits a discussion in main paper.

Response: The major difference between these two terms comes from the consideration of the electron cloud. The kinetic diameter (4.2 Angstrom) is defined as the diameter of the smallest cylinder that circumscribes the molecule based on the van der Waals radii of the atoms (*Phys. Chem. Chem. Phys.* **15**, 8795-8804 (2013); *J. Phys. Chem.* **97**, 1451-1454 (1993); *Zeolite molecular sieves: structure, chemistry, and use.* (Wiley, 1973)). Meanwhile, in our AIMD simulation the projected diameter (2.8 and 2.5 Angstrom) is the center-to-center distance of the atoms from the ends of the n-hexane and cyclohexane molecules, which unlike the van der Waals radii, does not consider the electron cloud of each atom. The interaction between the electron clouds of the atoms from the SWCNT and that of n-hexane were simulated separately in Supplementary Figure 8 when we determined the accessible pore size. To clarify this, we have noted that the projected diameter in the revised simulation methods refers to the atom center-to-center distance to differentiate it from the kinetic diameter:

“Note that: all the distances calculated in simulations are atom center-to-center distances.” (p. 20, lines 4-5)

We also clarify this in the caption of Supplementary Figure 10:

“Note: all the projected diameter calculated in our simulations are atom center-to-center distances.”

We also rephrased the following sentence to say:

“Because the curvature effects²⁷ become so pronounced in these small diameter nanotubes, the concept of van der Waals radii of atoms^{28,29}, which model atoms as hard spheres, and the kinetic diameters of molecules¹⁴, which are calculated based on the van der Waal radii, cannot fully capture the nanopore-molecule interactions” (p. 9, lines 12-14),

3. Figure 1 – relative sizes of the hexane, cyclohexane with respect to (6,5) nanotube. Nanotube diameter was 0.42 nm and the n-hexane kinetic diameter as mentioned in the paper was 0.43 nm. But the relative sizes as shown in the figure 1 seem way off with nanotube being shown as much smaller than the n-hexane. And coming to point 2 above, if the diameter is so low, it should be easily able to enter the nanotube.

Response: We thank the reviewer for pointing out this inconsistency, which arise from inconsistent scale for the two molecular structures. We have now corrected this in Figure 1.

4. In figure 1, the legend reads blue arrow represents emission – please correct this, as the bluish color in the figure is excitation.

Response: We thank the reviewer for pointing out this error. We have corrected the issue in the revised manuscript. The caption of Figure 1 now reads as: “The yellow arrow represents the PL emission from unfilled nanotube segments.”

5. According to the statement made in Page 5, second paragraph that “water filled which are expected to feature little to no strain since the molecular size of water is relatively small”, how would the authors explain the spectral shift of water-filled SWCNT from empty SWCNT shown in Figure 2e and Supplementary Figure 4 including the larger diameter ones which are not supposed to show a change as the diameter of water is small ? The authors need to amend the statements and explain clearly.

Response: The PL shift of SWCNTs caused by water filling has been studied and well understood in previous publications by others and one of us (*ACS Nano* **6**, 2649-55 (2012); *Angew. Chem. Int. Ed. Engl.* **50**, 2764-8 (2011); *ACS Nano* **5**, 3943-53 (2011), etc.). These studies found that the observed PL shift is due to the change in the dielectric environment from a vacuum (end-capped, empty) to water, which has a dielectric constant of ~80 at room temperature. Therefore, even for a large diameter nanotube, the PL shifts upon water-filling, but this is not due to strain and does not exhibit the mod dependence as with large molecules due to strain. To clarify this point, we have added the following statement with reference to the above citations to the revised manuscript:

“We can differentiate these effects of strain vs. the dielectric environment by comparing n-hexane, cyclohexane, and water-filled (~~which are expected to feature little to no strain since the molecular size of water is relatively small~~) nanotubes of different mod and relatively larger pore size. We note that water filling also causes a spectral shift due to the dielectric effect, but it features little to no strain since the molecular size of water is relatively small.^{26,27}” (p. 6, line 5-9)

6. It needs to be explained why one nanotube mod can cause a redshift while the other one mod will cause a blueshift. In page 5, it reads “radial strain-induced electronic effect”. However

since this is the whole basis of the measurement, it could be explained in more detail.

Response: We thank the reviewer for this suggestion. The theory was first proposed and studied by Yang et al. (*Phys. Rev. Lett.* **85**, 154-7 (2000)). Later Leeuw et al. (*Nano Lett.* **8**, 826-31 (2008)) stretched water-filled SWCNTs within a polymer composite in order to apply strain on the nanotubes and they experimentally observed this same mod-dependent PL shift. This strain-induced effect is caused by the Fermi point of the nanotubes being pushed away or moved closer to the Brillouin zone vertices, which is dependent on the mod of the SWCNTs. We have cited these studies in the manuscript to explain the mod-dependent PL shift upon the application of strain (ref 20 and 25, p. 6, line 2). To make this point clearer, we have added the following description to the text:

“In particular, the Fermi point is pushed away from or closer to the Brillouin zone vertices due to expansive strain depending on the mod of the specific SWCNT species.” (p. 6, lines 17-18)

7. Figure 4 – Figure 4a gives same information as in figure 1a and b, but a bit more clearly. The repetition can be avoided but make the figure 1a and b bit more clear (comment 4 above).

Response: In Figure 1, we provide a general picture that both molecules are larger than the SWCNT pore size. Figure 4a is meant to illustrate the concept of using the SWCNT to selectively remove n-hexane from cyclohexane after we had established the ability of the n-hexane to deform and enter the nanotube pore. Therefore, while the figures may appear similar, they in fact serve two separate purposes in the text.

8. The authors have introduced an accessible pore size which is used as a cut-off value for the probability evaluation. In the main context they are saying the value is about 0.269 nm, which is smaller than the size of cyclohexane with any configurations. In Page 8, it seems that the authors use the n-hexane filled SWCNT system to calculate the project radical diameter first, then derive the accessible pore size. If so, how can you use this value which is acquired from the n-hexane system to evaluate the transport of cyclohexane molecules? When you calculate the project radical diameter, the interaction of n-hexane with the SWCNT has been taken into consideration. It will be invalid when you examine a different molecule.

Response: We calculated the accessible pore size from ab-initio molecular dynamics simulations by placing the n-hexane molecule inside the SWCNT and then calculating the distribution of the minimum distance between the carbon of the SWCNT and the hydrogen of n-hexane, as shown in Supplementary Figure 9a. Twice this distance is then subtracted from the diameter of the SWCNT to obtain the accessible pore size.

Since the type of interactions between the hydrogen of cyclohexane and carbon of the SWCNTs are similar to that between the hydrogen of n-hexane and carbon of the SWCNTs (i.e., non-polar interactions between the sp^2 carbon of the SWCNT and the hydrogen atoms of the sp^3 carbon of n-hexane and cyclohexane), we conclude that the closest distance between the SWCNT carbon and hydrogen is the same for cyclohexane, therefore the accessible pore size should be the same. This accessible pore size would change if the type of interactions between the SWCNT wall and the molecule were different (e.g., benzene, non-polar interactions between the sp^2 carbon of the SWCNT and the hydrogen atoms of the sp^2

carbon of benzene).

Figure 5 – it is probably hard to read the message of the figure here. Let us take cyclohexane – the probability at the peak of cyclohexane is close to 5%, whereas the legend reads, “there is zero probability for cyclohexane to exist at a molecular configuration smaller than the pore size of (6,5)- SWCNTs.”. Can this be explained?

Response: The y-axis of Figure 5 is the probability that the molecular configuration exists in bulk (i.e., pure n-hexane or pure cyclohexane solution). The pore size of the (6,5)-SWCNT is indicated by Φ of 0.269 nm and a dot-dashed line in Figure 5; that is, when any molecule has a size smaller than this, it can enter the nanotube. If we look at the curve of cyclohexane on Figure 5, we can see that when the size of cyclohexane (x-axis) is smaller than 0.269 nm, the probability (y-axis) that it can exist in bulk is nearly zero, no matter how it reconfigures. The most probable (5%) cyclohexane configuration has a size of ~0.40 nm. Therefore, cyclohexane in general cannot enter the 0.269 nm pore.

To avoid confusion, we have changed the y-axis title to “probability in bulk” in Figure 5. Also, we emphasize Φ in the Figure 5 caption:

“The n-hexane molecule has 2.7 % probability at the 11.2 % (extensionally) stretched configuration (minimum projected diameter 0.251 nm) which is smaller than the accessible pore size of the (6,5)-SWCNT ($\Phi = 0.269$ nm).”

9. Figure 2 – shows the optical images of the nanotubes, along with the spectral images. SEM and/or images of the nanotubes are needed to provide the information of the morphology of the nanotubes

Response: It is difficult to image the morphology change of these SWCNTs using SEM since individualized (6,5)-SWCNTs have a sub-nanometer diameter and are surfactant coated. We have attempted a TEM study but the resulting images are smeared out by the surfactant (Figure R4). Additionally, to measure the filled nanotube morphology with TEM or SEM, we would ideally want to image the same nanotubes that had been characterized by PL imaging. However, the surface of the substrate we used for PL imaging is coated with insulating polystyrene (to prevent PL quenching). This substrate is not suitable for either SEM or TEM, which require a conducting substrate.

However, it is important to point out that the global morphology of each individual nanotube is resolved to some extent with our single tube hyperspectral imaging experiments, as presented in this work.

Figure R4. A TEM image of (6,5)-SWCNTs.

10. Does the filling of the nanotube change the nanotube topography? AFM images before and after the filling could hold a key to this information.

Response: Unfortunately, the expansion/contraction of the nanotube upon molecular filling is beyond the AFM resolution to our knowledge. Streit et al. (*J. Phys. Chem. C* 122, 11577–11585 (2018)) found the strain when nanotubes are filled with tetracosane is as low as 0.1%. In the case of (6,5)-SWCNTs, the resulting change in height (diameter) of the nanotube can be calculated to be ~ 0.001 nm based on the equation $\sigma = \frac{\Delta h}{h}$, where σ is the strain, h is the original diameter, and Δh is the change in diameter ([https://en.wikipedia.org/wiki/Deformation_\(physics\)](https://en.wikipedia.org/wiki/Deformation_(physics))). This change in height is beyond the commercial AFM vertical resolution (~ 0.01 nm). Additionally, in our studies the SWCNTs are embedded in surfactant (sodium deoxycholate), which is softer than the nanotubes and difficult to completely remove. This makes it even more difficult to accurately determine nanotube topography change as a result of molecular filling by AFM. We are pleased that PL imaging, as shown in our work, starts to reveal molecular insights that are otherwise difficult to capture, and it would be exciting if high resolution imaging techniques such as AFM and EM will be advanced to uncover further details.

Reviewer #3 (Remarks to the Author):

In this combined experimental/computational study, Qu and co-workers investigate the potential of several single-walled carbon nanotubes (SWCNTs) to separate n-hexane from cyclohexane based on the SWCNT's size selectivity. Using the photoluminescence shift of the first electronic transition upon adsorption, they conclusively demonstrate that n-hexane and cyclohexane can both be adsorbed inside all nanotubes except for the (6,5)-SWCNT, in which only n-hexane is adsorbed. This observation, which is deemed counterintuitive by the authors as both cyclohexane and n-hexane have larger kinetic diameters (KDs) than the van der Waals (vdW) pore size of the (6,5)-SWCNT, is explained using both classical and ab initio molecular dynamics (MD) simulations. These simulations point towards the ability of n-hexane to stretch and hence reduce its effective diameter to fit into the SWCNT. As cyclohexane is observed not to undergo a sufficient stretching, it is completely excluded from the nanotube, resulting in the selective filling of n-hexane mentioned in the title.

While this study conclusively shows the ability of (6,5)-SWCNT to sieve n-hexane from cyclohexane, the authors' focus on the vdW pore size and the KDs of the molecules to yield the apparent contradictory adsorption of n-hexane in (6,5)-SWCNTs is a very static view of molecular sieving. In the last decade, this static view has been continuously challenged, as both the vdW pore sizes and the KDs of molecules are inherently dynamic parameters that change over time and are influenced by external conditions such as temperature, pressure, and the solvent in which the molecules and SWCNTs are present. This is true for all materials, especially if they allow for substantial flexibility (see for instance 10.1021/jacs.7b01688). This flexibility, in this case exhibited both by the adsorbed molecules and the strained SWCNTs, is also demonstrated here in Figure 5: while the projected diameter is a property that shows large fluctuations and is characterized by a broad distribution, the KD of each molecule represents only a single value that does not take into account this broad distribution. As a result, the comparison between the KDs and the vdW pore size in the introduction that leads to the apparent contradiction is rather artificial.

Based on this assessment, the novelty of the manuscript is the successful sieving of n-hexane from cyclohexane in (6,5)-SWCNTs, which is successfully demonstrated both on experimental and computational grounds. In contrast, the apparent incompatibility between the observed n-hexane adsorption and the vdW pore size of the SWCNT that is smaller than the KD of n-hexane, an incompatibility that is emphasized even in the abstract, is artificial. I would therefore urge the authors to solely focus on the successful sieving properties of the (6,5)-SWCNT and the explanation from their MD simulations as shown in Fig. 5. However, this also limits the novelty of this manuscript and I'm therefore unsure whether it would be a good fit for the broad audience of Nature Communications. Below, more detailed suggestions for improvement are given that the authors may wish to take into account upon revision.

Response: We thank the reviewer for their very helpful comments. We agree that KD is a static view of molecules. Indeed, the cross-sectional diameter corresponding to the KD is calculated from the van der Waal radii of atoms and bond angles when the molecule is static (*Phys. Chem. Chem. Phys.* 15, 8795-8804 (2013); *J. Phys. Chem.* 97, 1451-1454 (1993);

Zeolite molecular sieves: structure, chemistry, and use. (Wiley, 1973)). While a static representation is to some degree “artificial”, KD is a concept that has been widely used to represent the molecular size in the field of molecular sieving (*Angew. Chem. Int. Ed.* **55**, 2048 (2016); *Chem. Soc. Rev.* **43**, 4470-4493 (2014); *Nat. Mater.* **17**, 1128–1133 (2018); *Micropor. Mesopor. Mat.* **292**, 109748 (2020); *J. Membr. Sci.* **593**, 117427 (2020); *J. Membr. Sci.* **585**, 1-9 (2019); *Chem. Mater.* **32**, 3715-3722. (2020); *Chem. Commun.* **52**, 443-452 (2016)). We agree that this concept of KD has its limitations; rather, we introduce the “minimal projected diameter” to better describes such dynamic molecular systems. Indeed, this is an insight that we intended to elucidate in our work through a combined experimental and theoretical effort. We have revised our introduction to clarify this point:

“This is despite the fact that plausible filling configurations of both molecules suggest that neither molecule should be able to enter the rigid pore” (p. 3, line 5)

We also clarified the need for comparison of the dynamic molecular size in the discussion, as we agree it is a better description of this system:

“Discussions

KD is a concept that has been widely used to represent the molecular size in the field of molecular sieving.³⁴⁻³⁶ Our work, however, provides an example that this static view of molecular size fails to capture the flexible nature of molecules in entering a tight space. The structural rigidity of SWCNTs allows us to study the foundation of such observation by isolating the flexibility of the adsorbate (n-hexane or cyclohexane in our case) from the pore deformation which could occur with flexible pores such as metal organic frameworks due to the structural motions of ligands and lower Young’s modulus³⁷. Instead of using the KD, we simulate the dynamic changes of the molecules in solution to obtain the minimal projected diameter, which can better describe such a molecular sieving phenomenon.” (p. 10-11, lines 18-6)

We also want to point out the degree of flexibility (i.e., how much the molecule can deform) can vary significantly among materials. The reviewer refers to a study on the flexibility of metal organic frameworks (MOFs) and how the flexibility of the pore material can change the selectivity of MOFs (10.1021/jacs.7b01688), implying that for this reason our observation of n-hexane entering the SWCNT is by itself not particularly novel. However, there are two distinct differences here: (1) In comparison to MOFs, SWCNTs are in fact extremely rigid and difficult to deform as a result of their conjugated lattice of sp² carbon atoms; (2) A key insight derived from our study is on the flexibility of the filling molecules, not on the pore flexibility.

The atomic motion (e.g., rotation of ligands) that is typical of the MOF system, is impossible for SWCNTs given their unique fully-conjugated structure. This rigidity is also underscored by the (6,5)-SWCNT Young’s modulus, which is > 1 TPa (*J. Appl. Phys.* **84**, 1939-1943 (1998)). In contrast, MOFs generally have a lower Young’s modulus < 20 Gpa (*Chem. Sci.* **10**, 10666-10679 (2019); *Chem. Soc. Rev.* **40**, 1059-80 (2011); *CrystEngComm.* **17**, 286-289 (2015)) and are commonly constructed with coordinate bonds and single bonds, which enable the observed deformability/flexibility. Similarly, the Young’s modulus of zeolites (another material commonly used in molecular sieving) is generally smaller (< 100 Gpa, *Angew. Chem. Int. Ed.* **45**, 38, 6329-6332 (2006)) than that of SWCNTs.

Meanwhile, the flexibility of the molecular adsorbate is generally not discussed in these

MOF and zeolite systems (e.g. *J. Phys. Chem. Lett.* **3**, 2130–2134 (2012); *Science* **358**, 1068–1071 (2017); and *Nat. Mater.* **17**, 1128–1133 (2018)), creating a knowledge gap in our understanding of how molecules enter nano-confined spaces. Our system is unique in that we are able to isolate the effect of the molecular adsorbate's flexibility in relation to molecular filling events due to the rigidity of the pore. As a result, we believe the novelty of our work is strong given we have demonstrated a system that can provide well-defined, rigid nanopores for studying new aspects in important processes, such as molecular sieving and molecular confinement in porous materials.

To clarify this point, we have added the Young's modulus of the nanotube in the manuscript to emphasize the rigidity of the pores:

“The SWCNT nanopores are chemically inert, structurally rigid (Young's modulus > 1 TPa), and atomically smooth cylinders, featuring inflexible pores that are well-defined by the nanotube cylinder that is constructed from a conjugated sp^2 carbon lattice. Additionally, these nanopores are tunable in size within the sub-nm range based on their individual atomic structure (*i.e.*, nanotube chirality, as defined by a pair of integers (n,m)). ” (p. 2, lines 17–21)

Furthermore, our work is the first demonstration of molecular sieving by SWCNTs. We believe it is also novel to experimentally observe the PL response induced by the encapsulated molecules (adsorbate) inside the rigid porous material (adsorbent) using our unique hyperspectral imaging capabilities. This allows us to track the encapsulated molecules on an individual SWCNT level instead of an ensemble average that has led to greater insights into structural effects. Our measurement technique cannot be experimentally done with MOFs or zeolite materials due to these compounds' lack of such a PL response that is intrinsic to SWCNTs. To further emphasize this novelty, we have added a line in the manuscript to highlight the PL method we used.

“By capturing the optical response of the nanotube at its first electronic transition (E_{11}) using both ensemble measurements and hyperspectral imaging of individual nanotubes, we find it is possible to study this molecular filling process at unprecedented detail to uncover a molecular level insights into the nanopore selectivity, as shown in the schematic of Figure 1c.” (p. 3, lines 10-14)

For these reasons, we respectfully disagree with the reviewer that our study lacks the novelty that would make it appropriate for *Nature Communications*. Our demonstration of the flexible deformation of a molecule entering a rigid, well-defined, and smaller-sized nanopore is a significant finding that we believe further advances fundamental knowledge of molecular sieving and other filtration processes.

1. Since the reported vdW pore sizes of the SWCNTs and the KDs of the adsorbed molecules play a crucial role in the current version of the manuscript, it would be instrumental to verify that the procedure leading to the vdW pore sizes of Table S1 is a sensible one. I doubt whether the geometry of graphite is representative for the interactions present in the strongly curved geometry of the SWCNTs and hence whether the vdW radii can be readily extracted using this

procedure. This is especially relevant since the observed difference between the n-hexane KD (0.43 nm) and the (6,5)-SWCNT vdW pore size (0.422 nm) is minimal, possibly invalidating this claim when slightly altering the procedure to determine the vdW pore size. In this respect, it would also be instructive to report the KD of cyclohexane, as well as the procedure with which these KDs are determined.

Response: We thank the reviewer for pointing out this oversight of ours. We have added the KD values of n-hexane and cyclohexane in the revised manuscript in the following sentence:

“Our series of experiments demonstrate that n-hexane (KD ~0.43 nm)^{13,14} is able to enter (6,5)-SWCNTs with a van der Waals pore size of just 0.422 nm, while cyclohexane (KD ~0.60 nm)^{14,15} is excluded.” (p. 3, line 3)

Both the KD of n-hexane and cyclohexane were obtained from the literature (*Phys. Chem. Chem. Phys.* 15, 8795-8804 (2013); *J. Phys. Chem.* 97, 1451-1454 (1993); *Zeolite molecular sieves: structure, chemistry, and use.* (Wiley, 1973)). The KD reported are defined as the diameter of the smallest cylinder that circumscribes the molecule based on the van der Waals radii.

In terms of the accuracy of the nanopore size, we agree with the reviewer’s speculation that curvature may play a role. However, unfortunately this is an open question as the nanotube van der Waals pore size is extremely difficult to measure accurately by experimental means, being considerably smaller than the instrument resolution available (e.g., electron microscopy and AFM). Given the limitation, to provide a basic conceptual point of view, the research community commonly uses the thickness of a layer of graphene to determine the van der Waals pore size of SWCNTs (e.g., *Nano Lett.* 17, 805-810 (2017); *J. Phys. Chem. C* 122, 11577-11585 (2018)). Therefore, in our initial assessments we used these common measures to compare the relative sizes of the molecules and the pores. In our work, we ultimately simulated these materials to more accurately calculate the accessible pore size. In effect, we used the literature’s best information to initially compare the different molecular sizes, but the novelty of our work rests in both the experimental and simulated observation that the slightly larger n-hexane molecule is able to deform in order to enter the smaller, rigid pore size of the nanotube – which has never been previously observed and should not be possible given prior knowledge.

To clarify the relative comparison of the molecular sizes and the limitations of the current knowledge, we have rephrased the following sentence:

“Because the curvature effects²⁷ become so pronounced in these small diameter nanotubes, the concept of van der Waals radii of atoms^{28,29}, which model atoms as hard spheres, and the kinetic diameters of molecules¹⁴, which were calculated based on van der Waal radii, cannot fully capture the nanopore-molecule interactions” (p. 9, lines 11-15)

In effect, we are in agreement with the reviewer on this important point, and believe we are providing further contribution to the argument that KD is not an appropriate description when discussing molecular filling and sieving events.

Finally, how are the vdW pore sizes of the SWCNTs influenced by the different experimental steps (thermal oxidation, incubation in solution, filtration)? Given the observation of strain and a PL shift upon adsorption, the altered interactions between the SWCNT and the adsorbed molecules will most likely also change the vdW pore size.

Response: Among the experimental steps, only the thermal oxidation alters the structure of the nanotubes by opening the nanotube ends, while the other steps the reviewer describes should have negligible effect on changing the van der Waals pore size. The nanotubes are synthesized with closed, capped structures at their ends. Due to the much higher curvature, these ends can be selectively burned away during thermal oxidation compared to the central (i.e., sidewall) locations of the nanotubes. This fact has been well-established in the literature (*Chem. Phys. Lett.* **386**, 239-243 (2004)), and is consistently supported by our Raman spectroscopy studies (see the added Supplementary Note 3 and Supplementary Figure 1 in the Supplementary Information). Supplementary Figure 1a shows the D/G ratio barely increases from 0.021 to 0.032 after this oxidative treatment, which indicates the sidewalls of the nanotubes have not been significantly modified. Additionally, our observation of bright PL from these nanotubes further indicates oxidized defects, if any, are negligible as they would have otherwise quench the PL (*Nano Lett.* **11**, 1636-1640 (2011)). Since these sidewalls determine the nanotube channel, the corresponding pore size should also remain unchanged. Although the size of pore mouth (two ends) may be slightly altered than the pore defined by the channel due to oxidation, we have shown from hyperspectral imaging that the n-hexane molecules can enter the center region of long (> 3 μm) nanotubes, the diameter of which is not affected by the pore mouth.

Additionally, the diameter change during incubation should be very small because the thermal expansion coefficient for the nanotube diameter change is as small as $(-0.15 \pm 0.20) \times 10^{-5} \text{ K}^{-1}$ (*Phys. Rev. B* **64**, 241402 (2001)). At the highest temperature (55 $^{\circ}\text{C}$) we used for incubation, the thermal expansion of (6,5)-SWCNTs with a carbon center-to-center diameter of 0.757 nm in the radial direction can be calculated by $\frac{\Delta h}{h} = \alpha \Delta T$ (where Δh is the change of diameter, h is the diameter of the nanotubes, α is the thermal expansion coefficient, and ΔT is the temperature change) to be extremely low at $(-3.4 \pm 4.5) \times 10^{-5}$ nm, which means the nanotube diameter is insensitive to the temperature. Therefore, the thermal expansion of the diameter should be negligible within the temperature range used during incubation.

Furthermore, because of the high Young's modulus and conjugated sp^2 carbon lattice of SWCNTs, they are not easy to deform upon the application of external force. Therefore, we have no reason to believe such a rigid material would be able to change its diameter during filtration.

Finally, we wanted to point out that we did not focus on studying the pore size of the SWCNTs after they absorbed n-hexane. However, Streit et al. (*J. Phys. Chem. C* **122**, 11577-11585 (2018)) found the strain when nanotubes are filled with tetracosane is as low as 0.1%. In the case of (6,5)-SWCNTs, the change in diameter of the nanotube can be calculated to be ~ 0.001 nm based on the equation $\sigma = \frac{\Delta h}{h}$, where σ is the strain, h is the original diameter, and Δh is the change in diameter

([https://en.wikipedia.org/wiki/Deformation_\(physics\)](https://en.wikipedia.org/wiki/Deformation_(physics))).

2. The statement on page 6 that “cyclohexane and n-hexane are nearly identical in structure” seems misplaced or insufficiently precise, given the substantial difference in their geometry.

Response: We thank the reviewer for pointing out this. We meant to convey their similarity in terms of “chemical identity,” as we have modified on the text page 7, line 14.

3. How long were the SWCNT present in the n-hexane and cyclohexane solutions before they were removed (see first paragraph of page 7)? Does this incubation time change the sieving properties?

Response: We have stated in the methods section that the SWCNTs were incubated in the solutions for 7 days in the molecular sieving experiment. We did not observe the incubation time to affect the kind of molecules that can enter (i.e., no matter how long we incubated in cyclohexane, we observed no evidence of cyclohexane entering (6,5)-nanotube). Herein, we provide a general way to fill the nanotubes. We give the molecules enough time to move, reconfigure, and enter the pore because the molecular reconfigurations, stretching, and diffusion for a n-hexane to enter the SWCNT pore occur spontaneously at room temperature and these motions happen on a sub-ms time scale, many orders of magnitude smaller than the contacting time. Campo et al. has demonstrated that alkanes with number of carbon atoms from 12 to 40 can readily fill SWCNTs within roughly 24 h (*Nanoscale Horiz.*, **1**, 317-324 (2016)). From the simulation (Supplementary Figure 11), we have demonstrated that n-hexane can enter the nanotube within 2 ps at 300 K, which is orders of magnitude shorter than our incubation time. Also, Figure R5 shows the PL spectra of n-hexane-filled (6,5)-SWCNTs incubated in different conditions. It clearly shows that incubation at 1 day at either room temperature or 55 °C allows n-hexane to readily fill (6,5)-SWCNTs. Longer incubation time does not affect the filling properties and resulting PL wavelength.

Figure R5. PL spectra of n-hexane-filled (6,5)-SWCNTs with different incubation conditions. Orange: incubated in n-hexane for 1 day at 55 °C, blue: incubated in n-hexane for 1 day at room temperature, red: incubated in n-hexane for 3 days at room temperature, black: incubated in n-hexane for 7 days at room temperature.

4. At the end of the first paragraph of page 7, the authors suggest that “extrapolation of the experimental curve in Figure 4b suggests that 24.1 mg/mL of end-opened SWCNTs can completely strip the n-hexane from the mixture to attain 100.0% pure cyclohexane.” This linear extrapolation of Figure 4b from 99.95% to 100% purity is not warranted based on the available data, which seem to indicate a flattening of the purity upon higher SWCNT concentration. If retained, this claim of 100% purity should also be experimentally validated.

Response: We thank the reviewer for this valuable suggestion. Due to the limited data, we cannot guarantee it is a linear relationship and will not flatten eventually. Therefore, we have deleted the claim below from the manuscript:

“Extrapolation of the experimental curve in Figure 4b suggests that 24.1 mg/mL of end-opened SWCNTs can completely strip the n-hexane from the mixture to attain 100.0 % pure cyclohexane.” (p. 8 line 13-15)

We have also added a note to clarify the use of trend lines in the Figure 4 caption:

“The two trend lines are added to guide the eye.”

5. On page 5 and in Figure 5, the authors illustrate the dynamic character of the projected diameter of the two molecules, which seems to be a better descriptor of the system than their static KDs. However, some questions remain on this part. What is the “relaxed state in the bulk solution” the authors refer to, and how is it determined? How were the stereoisomers determined to bin the MD generated molecules into – solely based on their projected diameter? How did the authors come to the 222,039 and 1,131,188 molecular configurations for n-hexane and cyclohexane, respectively? Can these molecular configurations be considered independent, or do they stem from a single simulation for each molecule? If they stem from a single simulation, it is necessary to check whether the observed distribution can be reproduced when using different initial conditions during the MD simulation. It would also be beneficial if the authors could provide geometry files for different relevant molecular configurations of n-hexane. Finally, the authors refer to “the stretched configuration of n-hexane” and an associated 2.7% probability to encounter it. Given the continuous character of the stretching observed in Figure 5, I would suggest to focus only on the probability of finding n-hexane with configurations that are smaller than the (6,5)-SWCNT’s accessible pore diameter. What does the blue shaded area in Fig. 5 represent?

Response: We thank the reviewer for the suggestions. The relaxed state in the bulk solution refers to the geometry that has the most probable projected width of n-hexane, which is 0.28 nm. The length of the n-hexane molecule at this projected width is 0.808 nm, as mentioned in Table S2. To clarify this point, we have revised the text as follows:

“Compared to the relaxed state (*i.e.*, the most probable configuration) in the bulk solution, the n-hexane molecule is stretched by 11.2 % (elongated along the length) inside the (6,5)-SWCNT pore (Table S2).” (p. 10, lines 4)

Since these calculations were done for the bulk solution, they can be considered

independent. The stereoisomers generated for binning were solely based on the projected diameter, since this is the most important parameter of the n-hexane molecule governing whether it enters (6,5)-SWCNTs or not. The numbers noted are the total number of configurations used in Fig. 5 to calculate the distribution of the projected width. The simulations were performed until smooth curves were obtained for the distribution. These configurations were collected from the simulations of n-hexane or cyclohexane placed in a bath of volume $2.5 \times 2.5 \times 2.5 \text{ nm}^3$ at 300 K and 1 atm. The simulations were equilibrated for 5 ns, before post-processing was done to bin the molecules based on their projected width. Additionally, the geometry files of the stretched and unstretched molecules of n-hexane will be uploaded as supplementary files.

The blue shaded area refers to the total probability of n-hexane entering the nanotubes, which we have clarified in the text:

“there is a 28.5 % total-overall probability (blue-shaded area in Figure 5)” (p. 10, lines 11–12).

Also, we clarify this in the caption of Figure 5: “The overall probability that any particular free n-hexane molecule exists in a conformation sufficiently narrow to enter the (6,5)-SWCNT interior, calculated through the ratio of the integral of the shaded blue area in the figure to the entire minimum projection curve, is 28.5 %.”

6. In the Methods section, the paragraph on “Molecular Dynamics (MD) and Ab initio Molecular Dynamics (AIMD)” is in its current form insufficient to reproduce the simulations and to validate whether the simulations are sufficiently accurate to back up the reported results. Below, I have listed seven such issues, but it may be expected more are present.

First, it is not always clear which parts of this section pertain to the force-field MD simulations and which to the AIMD simulations.

Response: The MD and AIMD sections have been separated and more detail has been added as follows:

“Molecular Dynamics (MD). MD simulations were performed using the Large-scale Atomic/Molecular Massively Parallel Simulator (LAMMPS)⁴¹ MD toolkit To simulate organic molecules in MD, we use the all-atom optimized potentials for liquid simulation (OPLS-AA) potential⁴⁴ to model carbon-carbon and carbon-hydrogen interactions. The atoms in the SWCNT are simulated using the Adaptive intermolecular reactive empirical bond order potential (AIREBO).⁴⁶ Interactions between carbon in the SWCNT and carbon and hydrogen in hexane and cyclohexane are modeled using Lennard-Jones potential⁴⁵ with parameters given in Table S3. The parameters for carbon and hydrogen were taken from ref. ⁴⁶ and oxygen from ref.⁴⁷ The SWCNTs are terminated with -OH groups, and the pressure is maintained at 0.1 MPa (1 atm) and temperature at 300 K to model the experimental conditions. MD simulations were performed for the cyclohexane, hexane, and (6,5)-SWCNT systems, shown in Supplementary Figure 8. The configurations to calculate the distribution of the projected width in Figure 5 were collected from the simulations of n-hexane/cyclohexane placed in a bath of volume $2.5 \times 2.5 \times 2.5 \text{ nm}^3$ at 300 K and 0.1 MPa (1 atm). The simulations are equilibrated for 5 ns, before post-processing is done to bin the molecules based on their

projected width. A time step of 0.25 fs was used, with data being sampled every 100 timesteps and the system was run until enough statistics were obtained to ensure a smooth distribution. The n-hexane system was run for ≈ 125 ps and the cyclohexane system for ≈ 600 ps (after equilibration of 5 ns using the canonical (NVT) ensemble) to obtain the required statistics.

Ab initio Molecular Dynamic Simulations (AIMD). AIMD simulations are performed using the Vienna Ab initio Simulation Package (VASP).⁴² The Perdew–Burke–Ernzerhof (PBE) functional⁴³ is used for exchange correlation energy. A $1 \times 1 \times 4$ grid has been used in these simulations along with an energy cut-off of 400 eV. Convergence with respect to the grid size and the energy cut-off is shown in Supplementary Figure 12. We calculate the geometry of the stretched molecule from MD simulations by relaxing all the angle parameters, allowing the molecule to freely stretch within the nanotube pore. AIMD simulations were performed for two sets of systems. In the first system (6,5)-SWCNT is simulated with n-hexane (both stretched and un-stretched) placed at the pore mouth to observe the molecules moving into the pore, as shown in Supplementary Figure 11. The second set of AIMD simulations are performed with stretched and un-stretched n-hexane placed inside the SWCNT, away from the hydroxyl groups at the pore mouth. This simulation is performed in order to calculate the accessible pore size. Each system has 1-unit cell of (6,5)-SWCNT consisting of 364 carbon atoms, with an additional 44 atoms of hydroxyl groups (-OH) at the end of the nanotube. In total, the systems have 428 atoms with the volume of the simulation box being $22.5 \times 22.5 \times 80 \text{ \AA}^3$. The data files for the coordinates of the systems used are supplied in the Supplementary File. All the geometries were initially relaxed with an energy convergence criterion of 10^{-6} eV. AIMD simulations were then performed on the relaxed structures until it was sufficient to determine whether the molecule was being excluded or not, which is ≈ 2 ps as shown in Supplementary Figure 11. Note that: all the distances calculated in simulations are based on atom center-to-center distances.” – p.18-19

Second, for the AIMD simulations, a $1 \times 1 \times 4$ k-point grid and an energy cut-off of 400 eV have been used. To verify whether these computational values are sufficient, it is important to confirm that the obtained results are converged with respect to the grid size and the energy cut-off, which is rather low. Especially the $1 \times 1 \times 4$ k-point grid seems odd, as the models of Fig. S10 are defined by one long direction (along the nanotube axis) and to shorter ones. As the k-point grid is defined in the reciprocal space, I would therefore expect a k-point grid of the type $1 \times 4 \times 4$, for instance.

Response: The convergence with respect to the grid-size and energy cut-off have been added to Supplementary Figure 12. Due to the single-file packing of water (of which the size is much smaller than that of n-hexane) inside (6,5)-SWCNTs (*Phys. Rev. Lett.* **118**, 027402 (2017)), the n-hexane molecules are expected to pack single file. Therefore, we only need to sample along the axis of the nanotubes. The $1 \times 1 \times 4$ provided indicated that we want to sample 4 points along the axis of the SWCNT (which is the third axis in the reciprocal space), and 1 point each in the other directions, as mentioned in documentation of VASP with the link (<https://www.vasp.at/wiki/index.php/KPOINTS>).

Third, how many atoms and molecules were present in these systems, and how large were the systems? Ideally, input files would be present for all relevant systems. This is especially relevant to reproduce the AIMD simulations with the stretched and unstretched configurations of n-hexane, as these descriptors do not fully determine the molecular configuration.

Response: The following additions have been made to the Methods section of the Molecular dynamics and AIMD in the revised manuscript

MD:

“The configurations to calculate the distribution of the projected width in Figure 5 were collected from the simulations of n-hexane/cyclohexane placed in a bath of volume $2.5 \times 2.5 \times 2.5 \text{ nm}^3$ at 300 K and 0.1 MPa (1 atm). The simulations were equilibrated for 5 ns, before post-processing is done to bin the molecules based on their projected width” (p. 18-19, lines 18-1)

AIMD:

“Each system has 1-unit cell of (6,5)-SWCNT consisting of 364 carbon atoms, with an additional 44 atoms of hydroxyl groups (-OH) at the end of the nanotube. In total, the systems have 428 atoms with the volume of the simulation box being $22.5 \times 22.5 \times 80 \text{ \AA}^3$. The data files for the coordinates of the systems used are supplied in the Supplementary File. All the geometries were initially relaxed with an energy convergence criterion of 10^{-6} eV. AIMD simulations were then performed on the relaxed structures until it was sufficient to determine whether the molecule was being excluded or not, which was ≈ 2 ps, as shown in Supplementary Figure 11.” (p. 19-20, lines 18-4)

Fourth, the molecules inside the nanotube were allowed to stretch, but only the angle parameters were relaxed. Why weren't the other internal parameters, such as bond lengths, relaxed, and why wasn't the nanotube allowed to relax?

Response: The nanotube was not allowed to relax (changing the bond and angle force constants) because the potential used for the nanotube, which is AIREBO, accounts for all the bond and angle fluctuation observed experimentally in SWCNTs and is extensively parameterized with experiments in the paper by Steven et al. (*J. Chem. Phys.* **112**, 6472-6486 (2000)). Furthermore, relaxing the other parameters like the bond and dihedral parameters did not have any additional effects on the entrance of n-hexanes to (6,5)-SWCNTs. That is why, in order to be as close to the real systems, only the force constants of the angles were reduced in order to account for the stretching.

Fifth, during the AIMD simulations, the molecules were placed at the pore mouth. How does the exact position and orientation of the molecules influence the adsorption or rejection of these molecules? In this paragraph, the authors should also refer to Fig. S10 instead of Fig. S9.

Response: We thank the reviewer for catching this error. The reference was changed from Fig S9 to Supplementary Figure 11 (p. 19, line 21) in the revised methods. With regards to the orientation, the axis of the n-hexane molecule must be parallel to the axis of the SWCNT, as the projected width of any other orientation becomes much larger than the accessible pore size of the SWCNT and this n-hexane is therefore excluded from the SWCNT. Furthermore, if the n-hexane molecule is completely encapsulated inside the SWCNT, the molecule does not

come out, rather stabilizing inside the SWCNT. Lastly, from MD simulations, it was observed that the n-hexane molecules can get to the orientation shown in Supplementary Figure 11a but are excluded if their projected width is larger than the accessible pore size of the SWCNT.

From the experimental point of view, because the mass ratio of n-hexane or cyclohexane to nanotubes used during the nanotube filling process was $\geq 66:1$, the number of n-hexane or cyclohexane molecules was in excess to the number of nanotubes. With enough time, one n-hexane can reconfigure randomly, and eventually enter the tubes when the size of n-hexane fits the pore. Simulation of an n-hexane with different orientation and different distance away from the nanotube can be done. However, it will consume a significant amount of computing time. We are approximating a nanotube in a continuous bath of n-hexane molecules, so one n-hexane will always be nearby the pore mouth. Thus, we think studying the effects of orientation or exact position of the molecules is beyond the scope of this study.

Sixth, how were the Lennard-Jones parameters of Table S3 determined – those are not discussed in Ref. 33? This is especially relevant since the Lennard-Jones parameters play a pivotal role during adsorption in tight pores – small changes in the sigma parameters can be expected to lead to substantial differences in the results of for instance Figure 5.

Response: The references for the Lennard-Jones parameters have been specified in the methods of the revised manuscript.

“The parameters for carbon and hydrogen were taken from ref.³⁹ and oxygen from ref.⁴⁰ (p. 18, line 15).

Finally, what were the thermostat and barostat relaxation times, and the total simulation times for all simulations performed in this manuscript?

Response: The details have been added to the MD and AIMD methods section.

“The simulations were equilibrated for 5 ns, before post-processing is done to bin the molecules based on their projected width. A time step of 0.25 fs was used, with data being sampled every 100 timesteps and the system was run until enough statistics were obtained to ensure a smooth distribution. The n-hexane system was run for ≈ 125 ps and the cyclohexane system for ≈ 600 ps (after equilibration of 5 ns using the canonical (NVT) ensemble) to obtain the required statistics.” (p. 18-19, lines 21-5)

“All the geometries were initially relaxed with an energy convergence criterion of 10^{-6} eV. AIMD simulations were then performed on the relaxed structures until it was sufficient to determine whether the molecule was being excluded or not, which is ≈ 2 ps as shown in Supplementary Figure 11.” (p. 20, lines 2-5)

7. A color bar is missing for Supplementary Figure 4, panes (a) and (b).

Response: Thanks for the note, we have added a color bar to Supplementary Figure 4a, b. (Supplementary Figure 5a, b in the revised manuscript).

REVIEWER COMMENTS

Reviewer #1 (Remarks to the Author):

The authors performed all suggested experiments and added their data and discussion in the manuscript or Supplementary Material. The revision has addressed all the points raised in my previous comments. I now recommend the publication of this work.

Reviewer #2 (Remarks to the Author):

The authors have answered the questions and have included recommended changes.

Reviewer #3 (Remarks to the Author):

Qu et al. have substantially revised and complemented their manuscript based on the suggestions raised by the reviewers, which is greatly appreciated. As a result, the revised manuscript is improved substantially. However, there are still some remaining remarks that were either not completely addressed in this revision or that emerged because the authors provided more methodological information (which was necessary to ensure reproducibility). I would therefore strongly recommend the authors to address these queries before publication.

1. Regarding the use of the kinetic diameter (KD) as a descriptor for molecular sieving (previously the introductory comment). The authors make a fair point that the flexibility of the system under study here stems from the adsorbed molecules, not from the adsorbent. In that sense, there is a clear distinction between the here reported results and previous observations of unexpected molecular sieving properties, for instance in zeolites, although the conclusion for both is the same: a static picture of either the adsorbate or the adsorbent is insufficient to describe molecular sieving when the sizes of both are similar. Since there is no fundamental “physical” surprise here, only a limitation of using the KD as a measure to predict sieving (as also mentioned by the authors), I would strongly encourage the authors to consistently mention this in their manuscript rather than stating that the molecules are too large to enter the pore. For instance, in the abstract, the authors mention “Here, we experimentally demonstrate the ability of SWCNTs with a vdW pore size of 0.42 nm to separate

n-hexane from cyclohexane-despite the fact that both molecules are larger than the rigid nanopore.” This is not correct; in fact, the authors explicitly show that n-hexane only adsorbs when its size is smaller than the vdW pore size of the SWCNT, as one would expect. This manuscript exactly illustrates that the static description (KD) to calculate the size of the molecules is insufficient and that a dynamic picture such as the one in Figure 5 is essential to predict molecular sieving (I would argue this is one of the main conclusions of the paper). A similar issue arises on lines 4-5 of page 3 (the plausible filling configurations only suggest neither molecule is able to adsorb *if* one uses the KD as a measure), and on lines 21-1 of pages 8-9.

2. Regarding the discussion and interpretation of Figure 5 (previously comment #5). I appreciate the authors adding the probability of finding an n-hexane conformation inside the blue shaded area (28.5%). However, the cited probability of 2.7% to find the n-hexane molecule at the 11.2% stretched configuration seems to be ill-defined. Conceptually, the probability of finding the n-hexane molecule at any given conformation will be zero, as necessary for a continuous probability distribution. The only reason a finite probability is found here, is that the authors use a binning procedure, which is indeed necessary to generate the distribution. However, this also means that the choice of bins and bin widths will strongly affect the 2.7% probability (but not the 28.5% probability). If the authors would for instance halve the bin width, the “probability to observe an n-hexane molecule at the 11.2% stretched configuration” would also decrease substantially. Therefore, I would argue not to mention this ill-defined 2.7% probability in the caption or in the text, but only the 28.5% probability of finding n-hexane in a conformation that allows it to be adsorbed inside the SWCNT. The latter probability is well-defined, as it is obtained by integrating over part of the probability distribution and therefore takes into account the bin width. The 2.7% probability is mentioned on lines 1-2 on page 2, on lines 12-13 on page 10, on line 12 on page 11, and on lines 8-10 on page 29.

3. Regarding the convergence plots for the k-point grid and energy cut-off (previously remark #6). I respectfully disagree with the authors that the 1x1x4 k-point grid is a sensible one. As mentioned in the VASP documentation (www.vasp.at/wiki/index.php/KPOINTS), section ‘Monkhorst-Pack’, the $N_1 \times N_2 \times N_3$ k-point grid for a tetragonal lattice is often chosen as $N_1:N_2:N_3 = 1/a_1 : 1/a_2 : 1/a_3$, with a_1 , a_2 , and a_3 the Bravais lattice vectors of the unit cell. As the authors use a $22.5 \times 22.5 \times 80 \text{ \AA}^3$ unit cell, this would imply a rule-of-thumb k-point grid of $4 \times 4 \times 1$, not $1 \times 1 \times 4$. I completely understand that the axis of the nanotube (the third axis) is the important (and therefore longer) direction in direct space, but this directly implies that the third axis is the one that should be sampled the least in reciprocal space. Therefore, while Supplementary Figure 12 clearly shows that $N_3 = 4$ yields converged results, the convergence of the N_1 and N_2 parameters should also be checked. The effect of increasing N_1 and N_2 will be larger than that of increasing N_3 . In addition, the cut-off energy seems rather low according to Supplementary Figure 12, as the difference between the energy at the cut-off energy and its converged value exceeds 5 eV.

4. Regarding the Lennard-Jones parameters of Supplementary Table 3 (previously remark #6). These values do not seem to coincide with the ones reported in ref. 43 or 45. For instance, ref. 43 reports a σ_{HH} value of 2.65 Å, while Supplementary Table 3 states 2.4 Å; ref. 45 reports a σ_{OO} value of 3.18 Å, Supplementary Table 3 states 3.166 Å. In addition, I would expect the O-H σ and ϵ parameters to be defined as the arithmetic and geometric mean of the corresponding O-O and H-H parameters, as in refs. 43 and 45. In Supplementary Table 3, however, σ_{OH} and σ_{OO} are the same but different from σ_{HH} , thereby not satisfying this mixing rule. Where does this deviation come from?

5. Regarding the thermostat and barostat settings (previously remark #6). The thermostat and barostat settings used in the MD simulation are, as far as I can tell, still not mentioned in the Methods section.

6. Given that the unstretched n-hexane molecule is too large for the SWCNT pore, how was the molecule placed inside the material for the AIMD simulation (lines 17-19 on page 19)?

Point-by-Point Response to Reviewers' Comments

(color codes: black – editorial and reviewers' original comments; blue – response; red – changes made to the revised manuscript)

REVIEWER COMMENTS

Reviewer #1 (Remarks to the Author):

The authors performed all suggested experiments and added their data and discussion in the manuscript or Supplementary Material. The revision has addressed all the points raised in my previous comments. I now recommend the publication of this work.

Response: We are pleased to know our revisions have addressed the reviewer' comments. Again, we thank the reviewer for his/her time and constructive inputs.

Reviewer #2 (Remarks to the Author):

The authors have answered the questions and have included recommended changes.

Response: We are pleased to know our revisions have addressed the reviewer' comments. Again, we thank the reviewer for his/her time and constructive inputs.

Reviewer #3 (Remarks to the Author):

Qu et al. have substantially revised and complemented their manuscript based on the suggestions raised by the reviewers, which is greatly appreciated. As a result, the revised manuscript is improved substantially. However, there are still some remaining remarks that were either not completely addressed in this revision or that emerged because the authors provided more methodological information (which was necessary to ensure reproducibility). I would therefore strongly recommend the authors to address these queries before publication.

1. Regarding the use of the kinetic diameter (KD) as a descriptor for molecular sieving (previously the introductory comment). The authors make a fair point that the flexibility of the system under study here stems from the adsorbed molecules, not from the adsorbent. In that sense, there is a clear distinction between the here reported results and previous observations of unexpected molecular sieving properties, for instance in zeolites, although the conclusion for both is the same: a static picture of either the adsorbate or the adsorbent is insufficient to describe molecular sieving when the sizes of both are similar. Since there is no fundamental "physical" surprise here, only a limitation of using the KD as a measure to predict sieving (as also mentioned by the authors), I would strongly encourage the authors to consistently mention this in their manuscript rather than stating that the molecules are too large to enter the pore. For instance, in the abstract, the authors mention "Here, we experimentally demonstrate the ability of SWCNTs with a vdW pore size of 0.42 nm to separate n-hexane from cyclohexane-despite the fact that both molecules are larger than the rigid nanopore." This is not correct; in fact, the authors explicitly show that n-hexane only adsorbs when its size is smaller than the vdW pore size of the SWCNT, as one would expect. This manuscript exactly illustrates that the static description (KD) to calculate the size of the molecules is

insufficient and that a dynamic picture such as the one in Figure 5 is essential to predict molecular sieving (I would argue this is one of the main conclusions of the paper). A similar issue arises on lines 4-5 of page 3 (the plausible filling configurations only suggest neither molecule is able to adsorb *if* one uses the KD as a measure), and on lines 21-1 of pages 8-9.

Response: We thank the reviewer for recognizing the novelty of our findings and suggesting a possible revision to avoid confusing. We have revised the manuscript as follow:

“Here, we experimentally demonstrate the ability of single-wall carbon nanotubes with a van der Waals pore size of 0.42 nm to separate n-hexane from cyclohexane—despite the fact that both molecules ~~have kinetic diameters are~~ larger than the rigid nanopore.”-abstract

“This is despite the fact that plausible filling configurations of both molecules suggest that neither molecule should be able to enter the rigid pore ~~when observed from the view of KD~~” – p.3 line 4-5

“~~However, the widely used concept of KD failed to capture this phenomenon this is surprising since the molecule is nominally larger than the cross-sectional opening of a (6,5)-SWCNT.~~” – p.8. line 17-19

2. Regarding the discussion and interpretation of Figure 5 (previously comment #5). I appreciate the authors adding the probability of finding an n-hexane conformation inside the blue shaded area (28.5%). However, the cited probability of 2.7% to find the n-hexane molecule at the 11.2% stretched configuration seems to be ill-defined. Conceptually, the probability of finding the n-hexane molecule at any given conformation will be zero, as necessary for a continuous probability distribution. The only reason a finite probability is found here, is that the authors use a binning procedure, which is indeed necessary to generate the distribution. However, this also means that the choice of bins and bin widths will strongly affect the 2.7% probability (but not the 28.5% probability). If the authors would for instance halve the bin width, the “probability to observe an n-hexane molecule at the 11.2% stretched configuration” would also decrease substantially. Therefore, I would argue not to mention this ill-defined 2.7% probability in the caption or in the text, but only the 28.5% probability of finding n-hexane in a conformation that allows it to be adsorbed inside the SWCNT. The latter probability is well-defined, as it is obtained by integrating over part of the probability distribution and therefore takes into account the bin width. The 2.7% probability is mentioned on lines 1-2 on page 2, on lines 12-13 on page 10, on line 12 on page 11, and on lines 8-10 on page 29.

Response: We thank for the valuable comment. We thus revised the manuscript as follow:

“Although at a relatively low ~~population probability (2.7-28.5 % overall)~~, ~~this the~~ stretched state of n-hexane does exist in the bulk solution, allowing the molecule to enter the tight pore even at room temperature.” -abstract

“Importantly, we find that the stretched configuration of n-hexane (with a minimum projected diameter of 0.251 nm) exists ~~with 2.7% probability~~ in the bulk solution at 300 K and overall there is a 28.5% overall probability (blue-shaded area in Figure 5) for a free n-hexane molecule to exist in molecular configurations that are smaller than the (6,5)-SWCNT’s accessible pore diameter (0.269 nm)” – p. 10 line 7-11

“MD simulations show that n-hexane enters the tight nanopore at its stretched state (elongating by nearly 11.2 %), which exists with an overall probability of 28.5% ~~2.7% population~~ in the bulk solution at 300 K.” – p.11 line 7-9

The n-hexane molecule ~~extensionally stretched has 2.7% probability at the 11.2% (extensionally) stretched configuration~~ with (a minimum projected diameter 0.251 nm) which is smaller than the accessible pore size of the (6,5)-SWCNT ($\Phi = 0.269$ nm). – Figure 5 caption

3. Regarding the convergence plots for the k-point grid and energy cut-off (previously remark #6). I respectfully disagree with the authors that the 1x1x4 k-point grid is a sensible one. As mentioned in the VASP documentation (www.vasp.at/wiki/index.php/KPOINTS), section ‘Monkhorst-Pack’, the $N_1 \times N_2 \times N_3$ k-point grid for a tetragonal lattice is often chosen as $N_1:N_2:N_3 = 1/a_1 : 1/a_2 : 1/a_3$, with a_1 , a_2 , and a_3 the Bravais lattice vectors of the unit cell. As the authors use a $22.5 \times 22.5 \times 80 \text{ \AA}^3$ unit cell, this would imply a rule-of-thumb k-point grid of $4 \times 4 \times 1$, not $1 \times 1 \times 4$. I completely understand that the axis of the nanotube (the third axis) is the important (and therefore longer) direction in direct space, but this directly implies that the third axis is the one that should be sampled the least in reciprocal space. Therefore, while Supplementary Figure 12 clearly shows that $N_3 = 4$ yields converged results, the convergence of the N_1 and N_2 parameters should also be checked. The effect of increasing N_1 and N_2 will be larger than that of increasing N_3 . In addition, the cut-off energy seems rather low according to Supplementary Figure 12, as the difference between the energy at the cut-off energy and its converged value exceeds 5 eV.

Response: It is indeed the case that according to the VASP documentation, a $4 \times 4 \times 1$ grid is recommended. In literature, though, for simulations involving CNTs, it is common to use a $1 \times 1 \times N$ grid (*J. Am. Chem. Soc.* **124**, 15076–15080 (2002); *Phys. Rev. B* **73**, 035413 (2006)), which was why we chose a $1 \times 1 \times 4$ grid.

To have more confidence on the inferences we have made from a $1 \times 1 \times 4$ grid, we performed additional simulations to compare the energies obtained from using a finer grid. We find that the energies obtained from the $4 \times 4 \times 1$, $2 \times 2 \times 2$ and $4 \times 4 \times 4$ grid differ from the values obtained from the $1 \times 1 \times 4$ by around 0.005 eV (Table R1). The downside of these finer grids is that they have higher number of k-points, making them computationally impractical and too expensive to perform AIMD simulations. This is because the number of atoms (N) in the system exceeds 400 and the computational cost scales as $\sim O(N^3)$. Since the energies used for the $1 \times 1 \times 4$ grid is close to the finer grids, we believe that the inferences obtained from AIMD simulations are reasonable.

Table R1. Convergence cut-off energy vs grid size

Grid size	Cut-off energy (eV)
1x1x4	-3963.3504
2x2x2	-3963.3556
4x4x1	-3963.3554
4x4x4	-3963.3562

It is true that the value used for cut-off energy (400 eV) is right on the border of where the plot in Supplementary Figure 12.b starts converging. The computational cost of running an AIMD simulation with 500 eV as the cut-off energy is ~40% larger when compared to running with 400 eV, which makes doing AIMD simulations impractical with a cut-off energy of 500 eV. This is the reason why we chose 400eV as the cut-off.

We have added the following note to the methods section in the manuscript

“Using 400 eV, the energy difference is 2.91 eV from the converged value, which occurs at a cutoff energy of 500 eV, but the latter is $\approx 40\%$ more expensive and makes performing the simulation impractical. For this reason, we chose a cutoff energy of 400 eV as a compromise for these AIMD simulations.”

4. Regarding the Lennard-Jones parameters of Supplementary Table 3 (previously remark #6). These values do not seem to coincide with the ones reported in ref. 43 or 45. For instance, ref. 43 reports a σ_{HH} value of 2.65 Å, while Supplementary Table 3 states 2.4 Å; ref. 45 reports a σ_{OO} value of 3.18 Å, Supplementary Table 3 states 3.166 Å. In addition, I would expect the O-H σ and ϵ parameters to be defined as the arithmetic and geometric mean of the corresponding O-O and H-H parameters, as in refs. 43 and 45. In Supplementary Table 3, however, σ_{OH} and σ_{OO} are the same but different from σ_{HH} , thereby not satisfying this mixing rule. Where does this deviation come from?

Response: We apologize for the confusion. The LJ parameters for oxygen come from SPC/E water model. The original paper (*Chem. Phys. Lett.* 294, 135-142 (1998)) has obscure units, which was why we had used ref. 45 (*J. Chem. Phys.* 119, 5185-5197 (2003).)), which has slightly different values. We have changed the reference to the original paper.

Further, in literature, H-H interactions are usually set to $\epsilon_{\text{HH}}=0$ and $\sigma_{\text{HH}} = 0$. Since we are considering tight-fitting molecules, we chose parameters which would better represent the system, the parameters used for H-H used from the reference (*Chem. Phys. Lett.* 294, 135-142 (1998)). We apologize for missing this out earlier. This reference is used as our system has a combination of hydrogen atoms attached to alkane/cycloalkane carbons and also those attached to oxygen atoms.

The parameters used for interaction between O and H are assumed to be similar to that of water, where interaction between hydrogen atoms is turned off ($\epsilon_{\text{HH}}=0$), which is why the same values for OH and OO have been used. We note there was a typo which we have now corrected in Suppl. Table 3 (Please see below).

Finally, it is found that n-hexane enters (6,5)-SWCNT in MD, only when very extreme, and unphysical values of $\sigma_{\text{C-H}}$ (<2.4 Å) and $\sigma_{\text{C-C}}$ (< 2.8 Å) are used. Cyclohexane would require even smaller values of $\sigma_{\text{C-H}}$ and $\sigma_{\text{C-C}}$. Altering σ values of other combinations has no effect on the movement of n-hexane/cyclohexane into (6,5)-SWCNT. If we change only $\epsilon_{\text{C-C}}/\epsilon_{\text{C-H}}$ without changing σ , then the extreme values of n-hexane movement into (6,5)-SWCNT is 0.01 eV, which is around 3 times the $\epsilon_{\text{C-C}}$ value used and therefore not physical.

We have revised the references in the MD method as following:

“The parameters for carbon were taken from ref.⁴³ and hydrogen were taken from ref.⁴⁵ and oxygen were taken from ref.⁴⁶” – p.17 line 9-11

Corrected Suppl Table 3, corrected value shown in red

Atom pairs	σ (Å)	ϵ (eV)
C-C	3.4	0.00286
C-O	3.024	0.0049
C-H	2.9	0.00192
O-H	3.166	0.00667
O-O	3.166	0.00867 0.00667
H-H	2.4	0.00129

5. Regarding the thermostat and barostat settings (previously remark #6). The thermostat and barostat settings used in the MD simulation are, as far as I can tell, still not mentioned in the Methods section.

Response: We apologize for missing that information. We have added the information to the methods section of MD.

“The damping parameters used in LAMMPS for the thermostat and barostat are 2.5 fs and 25 fs respectively under the NPT ensemble during production.” – p.17 line 19-20

6. Given that the unstretched n-hexane molecule is too large for the SWCNT pore, how was the molecule placed inside the material for the AIMD simulation (lines 17-19 on page 19)?

Response: If the van der Waals radii are not taken into consideration and only the atom center-to-center distances are considered (which determine the coordinates of the molecules), the coordinates of n-hexane do not overlap with those of (6,5)-SWCNT. After placing one n-hexane inside the SWCNT, we relax the system in order to minimize the energy of the given system. Finally, the AIMD simulation is performed, where, due to unfavorable free-energy of unstretched n-hexane, it moves out of (6,5)-SWCNT.

REVIEWER COMMENTS

Reviewer #3 (Remarks to the Author):

I thank Qu et al. for the further revisions and complementary information provided in their manuscript. Besides a few very minor remarks (see below), there is still one major comment that prevents me from recommending this article for publication. As per their answer to comment #4 in the previous round of revision, the authors obtained the Lennard-Jones (LJ) parameters from different sources (refs. 42, 44 and 45 in the manuscript). Unfortunately, because of this, the LJ parameters are not consistent, i.e., they do not satisfy a fixed mixing rule, in contrast to the original references they were taken from. This is known to affect the physical outcomes of the simulation (see, e.g., 10.1063/1.4867498). I'm therefore unsure whether the combination of LJ parameters used here makes sense from a physical point of view, and hence whether the outcomes reported here correspond with what should be expected from a physical system. Especially the choice to use the same LJ parameters for O-H and O-O pairs while having nonzero parameters for the H-H pair is troublesome.

Besides this major remark, there are few remaining minor remarks as given below:

1. In the force field MD part of the Methods section, the authors have changed in this revision “the canonical (NVT) ensemble” to “the canonical (NPT) ensemble”. This statement is now incorrect. It is either the canonical (NVT) ensemble, or the isothermal-isobaric (NPT) ensemble.
2. While the authors removed the discussion on Figure 4 regarding the extrapolation to a purity of 100%, the black and red lines in Figure 4 are still unnecessary extrapolated to a concentration of 25 mg/mL, while the data only runs to about 12 mg/mL. I would therefore suggest to also end the lines at 12 mg/mL, as the data presented here does not warrant such an extrapolation to 25 mg/mL.
3. I would encourage the authors to also take up the discussion on the grid size in their reply to comment #3 in the Supplementary Information, for instance when discussing the determination of the cut-off energy. Please note, however, that the second column of Table R1 does not denote the cut-off energy but the electronic energy of the system.

Point-by-Point Response to Reviewers' Comments

(color codes: black – editorial and reviewers' original comments; blue – response; red – changes made to the revised manuscript)

REVIEWER COMMENTS

Reviewer #3 (Remarks to the Author):

I thank Qu et al. for the further revisions and complementary information provided in their manuscript. Besides a few very minor remarks (see below), there is still one major comment that prevents me from recommending this article for publication. As per their answer to comment #4 in the previous round of revision, the authors obtained the Lennard-Jones (LJ) parameters from different sources (refs. 42, 44 and 45 in the manuscript). Unfortunately, because of this, the LJ parameters are not consistent, i.e., they do not satisfy a fixed mixing rule, in contrast to the original references they were taken from. This is known to affect the physical outcomes of the simulation (see, e.g., 10.1063/1.4867498). I'm therefore unsure whether the combination of LJ parameters used here makes sense from a physical point of view, and hence whether the outcomes reported here correspond with what should be expected from a physical system. Especially the choice to use the same LJ parameters for O-H and O-O pairs while having nonzero parameters for the H-H pair is troublesome.

Response: We thank the reviewer for the comment.

First, we would like to point out that the conclusion we make from MD simulations is the existence of a threshold diameter below which a cyclohexane molecule is excluded from entering the CNT but allowing an n-hexane molecule to pass through due to the difference in their molecular sizes (Suppl. Fig. 10). Since the curvature effects of small diameter CNTs become important and are not captured by classical MD, we opted to optimize the geometry at the quantum level using DFT and simulate of the movement of n-hexane into (6,5)-SWCNTs using AIMD. All the interactions in AIMD are calculated quantum-mechanically, therefore we do not need to use any LJ interaction parameters for these.

Next, we use non-zero parameters for the H-H pair as we wanted to incorporate the H-H interactions of the hydrogen atoms in n-hexane, which would otherwise be neglected if we used H-H parameters from the SPC/E model of water (where the H-H interactions are set to zero). In addition, using mixing rules for O-H interactions might not always be accurate as shown in the references below. Using mixing rules is an approximate way to estimate interactions between different types of atom. It is always better if we can use improved values with a justification or best, if we have full-fledged forcefield calculations for interactions. (*J. Phys. Chem. B*, 119 (2015), 5113-5123 and *Mol. Phys.*, 99 (2001), 619-625)

Finally, using Lorentz-Berthelot mixing rules does not affect our conclusion made from MD. To confirm this, we performed the same set of simulations using mixing rules (Table R1, Figure R1), checking whether n-hexane enters into (6,5)-SWCNTs or not. The plot for density of n-hexane is plotted as a function of the coordinate parallel to the axis of SWCNT. The setup for the system is described in Suppl. Fig. 8 (As we go from left to right in the plot, we go from a reservoir of n-hexane to the CNT and back to a reservoir of n-hexane). The results obtained from the new simulations are the same as

those before (i.e., n-hexane is excluded from the (6,5)-SWCNTs pore, as the density of n-hexane drops to zero between $z=27.5 \text{ \AA}$ and 67.5 \AA , which spans the coordinates of the CNT).

Table R1: Parameters that modified using L-B mixing rules (all remaining parameters remain the same)

Atom pairs	$\sigma(\text{\AA})$	$\epsilon(\text{eV})$
C-O	3.28	0.0044
O-H	2.78	0.0030

Figure R1. Comparison between the density of n-hexane in the system (Suppl. Fig. 8) when using mixing rules for O-H interactions and the current results. Density is plotted Vs the coordinate parallel to the CNT, which drops to zero between the coordinates 27.5 and 67.5 \AA , indicating the absence of n-hexane inside the CNT.

Besides this major remark, there are few remaining minor remarks as given below:

1. In the force field MD part of the Methods section, the authors have changed in this revision “the canonical (NVT) ensemble” to “the canonical (NPT) ensemble”. This statement is now incorrect. It is either the canonical (NVT) ensemble, or the isothermal-isobaric (NPT) ensemble.

Response: We have corrected this error which now reads “isothermal-isobaric (NPT) ensemble.”

2. While the authors removed the discussion on Figure 4 regarding the extrapolation to a purity of 100%, the black and red lines in Figure 4 are still unnecessary extrapolated to a concentration of 25

mg/mL, while the data only runs to about 12 mg/mL. I would therefore suggest to also end the lines at 12 mg/mL, as the data presented here does not warrant such an extrapolation to 25 mg/mL.

Response: We thank the reviewer for pointing this out. However, since the interior space of SWCNTs is linearly proportional to the mass of SWCNTs, the amount of adsorbed n-hexane should logically be proportional to the mass of SWCNTs. To avoid any confusion, we do emphasize in the figure caption that “The two trendlines are added to guide the eye **assuming a linear extrapolation.**”

3. I would encourage the authors to also take up the discussion on the grid size in their reply to comment #3 in the Supplementary Information, for instance when discussing the determination of the cut-off energy. Please note, however, that the second column of Table R1 does not denote the cut-off energy but the electronic energy of the system.

Response: We have added some additional discussion as supplementary note 4 on the grid size and supplementary table 4 to the SI

“Supplementary Note 4: Grid size used in simulation

To have more confidence on the inferences we might have made from the 1x1x4 grid, we performed additional simulations to compare the energies obtained from using a finer grid. We find that the energies obtained from the 4x4x1, 2x2x2 and 4x4x4 grid differ from the values obtained from the 1x1x4 by around 0.005 eV (Supplementary Table 4). These finer grids have higher number of k-points, making them computationally impractical and too expensive to perform AIMD simulations since the number of atoms (N) in the system exceeds 400 and the computational cost scales as $\sim O(N^3)$. However, the energies obtained from the 1x1x4 grid are close to those from the finer grids. We also note that the value used for the cut-off energy (400 eV) is right on the border of where the plot in Supplementary Figure 12.b starts converging. Using 400 eV, the error in the energy from its converged value of -3963.3562 eV is only 0.07%, which makes it a reasonable balance between precision and computing cost.”

Grid size	Total electronic energy (eV)
1x1x4	-3963.3504 eV
2x2x2	-3963.3556 eV
4x4x1	-3963.3554 eV
4x4x4	-3963.3562 eV

Supplementary Table 4. Grid size and corresponding total electronic energy (eV) in AIMD

The following statements were added to the caption of Supplementary figure 8 and Supplementary figure 10, respectively.

“Note that completely relaxed angle parameters are used for n-hexane in this simulation” – Supplementary figure 8 caption

“Geometries for n-hexane and cyclohexane are optimized using Density Functional Theory (DFT)” – Supplementary figure 10 caption

REVIEWER COMMENTS

Reviewer #3 (Remarks to the Author):

I thank Qu et al. for the further revisions and complementary information provided in their manuscript. Given that the remarks below are covered, I'm happy to recommend this revised manuscript for publication.

1. Both in their rebuttal and main text, the authors note that the force field MD simulations are only used to reveal the existence of a threshold pore diameter below which only n-hexane enters while cyclohexane cannot. Is this threshold pore diameter the one shown in Figure 3 as the boundary between the white and orange regions around 0.44 nm? If so, the authors may want to include in the caption that this threshold was obtained from MD simulations, as it was unclear to me that Figure 3 also included simulated data. However, if this is indeed the case, Figure 3 seems to be in contradiction with Figure R1, as the authors demonstrate in Figure R1 that n-hexane **does not** enter the (6,5)-SWCNTs, in contrast to the claim in the main text and Figure 3. If, on the other hand, the threshold diameter obtained from these MD simulations is not yet reported, it seems that this threshold diameter should be included in the text and compared to the actual pore diameters of the experimentally used SWCNTs. Which range of pore diameters was used in the MD simulations to obtain this results?

2. In the caption of Supplementary Figure 8, the authors note that "the completely relaxed angle parameters are used for n-hexane in this simulation". However, following the description of the MD simulations in the Methods section, it seems none of the atoms are assumed rigid. At what point are the completely relaxed angle parameters then used?

3. The procedure followed for the DFT optimizations, which are performed to obtain the optimized geometries for n-hexane and cyclohexane, as indicated in the caption of Supplementary Figure 10, is not reported in the Methods section. This includes the software, functional, optimization thresholds, and basis set that were used during optimization. Furthermore, it should be verified that the obtained minimum indeed leads to a molecule with only positive frequencies, although that shouldn't be a problem for these small molecules.

Point-by-Point Response to Reviewers' Comments

REVIEWER COMMENTS

Reviewer #3 (Remarks to the Author):

I thank Qu et al. for the further revisions and complementary information provided in their manuscript. Given that the remarks below are covered, I'm happy to recommend this revised manuscript for publication.

1. Both in their rebuttal and main text, the authors note that the force field MD simulations are only used to reveal the existence of a threshold pore diameter below which only n-hexane enters while cyclohexane cannot. Is this threshold pore diameter the one shown in Figure 3 as the boundary between the white and orange regions around 0.44 nm? If so, the authors may want to include in the caption that this threshold was obtained from MD simulations, as it was unclear to me that Figure 3 also included simulated data. However, if this is indeed the case, Figure 3 seems to be in contradiction with Figure R1, as the authors demonstrate in Figure R1 that n-hexane *does not* enter the (6,5)-SWCNTs, in contrast to the claim in the main text and Figure 3. If, on the other hand, the threshold diameter obtained from these MD simulations is not yet reported, it seems that this threshold diameter should be included in the text and compared to the actual pore diameters of the experimentally used SWCNTs. Which range of pore diameters was used in the MD simulations to obtain this results?

Response: We would like to clarify that Fig. 3 is from experimental observations and no data from MD simulations is used in that figure. The existence of a threshold pore diameter below which cyclohexane cannot enter can be inferred directly from our experimental observations of the filling behavior of cyclohexane and n-hexane as a function of the pore size, as shown in Fig. 3. To avoid confusion, we have revised the caption of Figure 3 to clarify this experimental origin. We note that the van der Waals pore size used in Figure 3 is based on wrapping a graphene sheet, as widely cited in the literature. We have included a note in the caption of Supplementary Table 1 to clarify the physical model that the van der Waals pore size of nanotubes is based on.

Revised caption for Figure 3:

“Figure 3. Molecular filling of nanotube pores is size dependent. The PL energy differences between cyclohexane (99.9%)- and n-hexane-incubated end-opened SWCNTs ($\Delta E_{11} = E_{11, \text{cyclohexane}} - E_{11, \text{n-hexane}}$) are obtained from experiments and plotted as a function of the van der Waals pore size of the nanotubes. Both n-hexane and cyclohexane can fit in those nanotube pores in the yellow colored area, while only n-hexane can enter the (6,5)-SWCNT pore, suggesting the existence of a threshold pore size below which n-hexane can enter but cyclohexane cannot. Note that the curves are added to guide the eye. The error bars represent the standard deviation of the E_{11} emission peak position measured from multiple different SWCNT samples. Uncertainty in the calculated points are represented by error bars equal to one standard deviation.”

Revised caption: **“Supplementary Table 1. van der Waals pore sizes of the nanotubes used**

* calculated by subtracting the van der Waals diameter of carbon atom from the carbon center-to-center diameter of SWCNTs. Note that the van der Waals diameter of carbon atom used here

is the interlayer spacing in graphite (0.335 nm). The diameter of (n,m)-SWCNT is calculated from $d_t = \frac{\sqrt{3}l_{C-C}}{\pi} \sqrt{n^2 + m^2 + mn}$ based on wrapping a graphene sheet with a C-C bond distance l_{C-C} of 1.44 Å.”

We note that MD simulations of molecular filling behavior corroborate experimental observations shown in Figure 3. In these simulations, we investigated the filling behavior for a range of CNTs in increasing order of diameter, i.e., (6,5)-, (8,3)-, (7,5)-, (8,4)-, (7,6)- and (9,4)-SWCNT (van der Waals pore size ranging from 0.422 nm to 0.581 nm). From the simulations, we can identify the threshold at which hexane enters whereas cyclohexane cannot. Both simulations and experimental observations (Fig 3) are in agreement; i.e., (8,3) CNT exhibits no selectivity (both n-hexane and cyclohexane are observed in CNT) and (6,5) CNT exhibits selectivity where only n-hexane is observed. However, these MD simulations do not give the pore size, which might have been a confusing point. We have added Supplementary Note 5, Supplementary Figure 13 and Supplementary Movies 3-6 in the SI to further clarify the points:

“Supplementary Note 5: Conformational changes of n-hexane in MD simulation

OPLS-AA potential⁶ is used to model n-hexane and cyclohexane, which determines the bond lengths, angles and dihedrals of these molecules. This potential allows for small deformations about the mean for bond lengths, angles and dihedrals. However, completely relaxing angle parameters (while keeping the bond and dihedral parameters the same as the OPLS-AA model) allows for large conformational changes in these molecules, thereby allowing them to freely stretch and compress. Only after relaxing the angle parameters, we observed that n-hexane molecules could enter (6,5)-SWCNT, with cyclohexane still being excluded and this result is consistent with AIMD calculations. To estimate the threshold at which the hexane enters whereas cyclohexane cannot, a range of SWCNTs in increasing order of diameter, i.e., (6,5)-, (8,3)-, (7,5)-, (8,4)-, (7,6)- and (9,4)-SWCNT ((van der Waals pore size ranging from 0.422 nm to 0.581 nm) were investigated. We then kept the SWCNT in an n-hexane/cyclohexane bath at 300 K, as shown in Supplementary Figure 8, and modeled the system as an isothermal-isobaric ensemble, in which the number of molecules, pressure, and temperature were held constant. The MD results corroborate our experimental observations that (8,3)-SWCNT and large diameter nanotubes exhibit no selectivity (both n-hexane and cyclohexane are observed to enter the SWCNT (Supplementary Figure 13 and Supplementary Movies 3-6) while (6,5)-SWCNT exhibits selectivity allowing only n-hexane to enter the nanopore.”

Finally, we would like to clarify that Fig. R1 from the previous reply compares the LJ parameters without accounting for large conformational changes (i.e. without relaxing angle parameters), to show that the mixing parameters for O-H do not change the results. The model in that response did not allow full relaxation of the angle parameters. For that reason, the results may look contradictory, but they are actually NOT. Indeed, if we plot the same figure, but now with the angle parameters relaxed to account for large conformational changes, we would again see that using mixing rules for O-H has the same results as those discussed in the paper. Please see Figure R2.

Figure R2. Comparison between the density of n-hexane in the system (Suppl. Fig. 8) when using mixing rules for O-H interactions and the current results using relaxed angle parameters. The three peaks between 27.5 Å and 67.5 Å indicate the presence of 3 molecules of n-hexane in a (6,5)-SWCNT of length 40 Å.

2. In the caption of Supplementary Figure 8, the authors note that “the completely relaxed angle parameters are used for n-hexane in this simulation”. However, following the description of the MD simulations in the Methods section, it seems none of the atoms are assumed rigid. At what point are the completely relaxed angle parameters then used?

Response: OPLS-AA potential (*J. Am. Chem. Soc.* 118, 11225-11236 (1996)) is used to model n-hexane and cyclohexane, which determines the bond lengths, angles and dihedrals of these molecules. This potential allows for small deformations about the mean for bond lengths, angles and dihedrals. Therefore, the angles/bonds/dihedrals are NOT frozen, and the molecules are NOT rigid. However, completely relaxing angle parameters (while keeping the bond and dihedral parameters the same as the OPLS-AA model) allows for large conformational changes in these molecules, thereby allowing them to freely stretch/compress. Only after relaxing the angle parameters, we observed that n-hexane molecules could move into (6,5)-SWCNT, with cyclohexane still being excluded and this result is consistent with AIMD calculations.

We have revised the Methods section of MD to clarify these points. Additionally, Supplementary Note 5 has been added to the SI to further clarify the points mentioned in questions 1 and 2 concerning MD simulations.

3. The procedure followed for the DFT optimizations, which are performed to obtain the optimized geometries for n-hexane and cyclohexane, as indicated in the caption of Supplementary Figure 10, is not reported in the Methods section. This includes the software, functional, optimization thresholds, and basis set that were used during optimization.

Furthermore, it should be verified that the obtained minimum indeed leads to a molecule with only positive frequencies, although that shouldn't be a problem for these small molecules.

Response: The procedure for the geometry optimization in DFT has been added to the Methods section, and the Methods section of AIMD was modified accordingly (as some of the information was made redundant with the addition of the new section on DFT).

These sections now read as:

“Density Functional Theory (DFT). DFT was used to optimize the geometries used in AIMD simulations (described in the next section) and to verify that the minimum projected diameter of n-hexane and cyclohexane in their relaxed state (most probable configuration) match with the results obtained from MD. These simulations are performed using the Vienna Ab initio Simulation Package (VASP)⁴⁶. The Perdew–Burke–Ernzerhof (PBE) functional⁴⁷ is used for exchange correlation energy. A $1 \times 1 \times 4$ grid has been used in these simulations along with an energy cut-off of 400 eV. Convergence with respect to the grid size and the energy cut-off is shown in Supplementary Figure 12. We note that using 400 eV, the energy difference is 2.91 eV from the converged value, which occurs at an energy cut-off of 500 eV, but the latter simulation is $\approx 40\%$ more expensive, which is why 400 eV was chosen as a compromise. It was verified that the optimization of the n-hexane and cyclohexane geometries leads to positive vibrational frequencies only and that the projected diameters of the most probable configurations of n-hexane and cyclohexane are in agreement with those obtained from MD (0.28 nm and 0.4 nm, respectively).

Ab initio Molecular Dynamic Simulations (AIMD). AIMD simulations were performed to show that n-hexane moves into (6,5)-SWCNT only when it is stretched (Supplementary Fig. 11) and to calculate the accessible pore size of (6,5)-SWCNT (Supplementary Fig. 9). These simulations were performed for the n-hexane and (6,5)-SWCNT systems starting from two configurations: (1) n-hexane (both stretched and unstretched) is placed at the pore mouth; (2) stretched n-hexane placed inside the SWCNT, away from the hydroxyl groups at the pore mouth. The modeled (6,5)-SWCNT has 1-unit cell consisting of 364 carbon atoms, with an additional 44 atoms of hydroxyl groups (-OH) at the end of the nanotube. In total, the systems have 428 atoms with the volume of the simulation box being $22.5 \times 22.5 \times 80 \text{ \AA}^3$. The data files for the coordinates of the systems used are supplied in the Supplementary File. The starting geometry of the stretched n-hexane molecule is calculated from MD simulations by relaxing all the angle parameters, allowing the molecule to freely stretch within the nanotube pore. AIMD simulations were performed in VASP and all the parameters used were same as those used for geometry optimization in DFT (mentioned in the previous section). For the system described in the first configuration, the simulations were performed until it was sufficient to determine whether the molecule was being excluded or not, which is ≈ 2 ps as shown in Supplementary Figure 11. ”

REVIEWERS' COMMENTS

Reviewer #3 (Remarks to the Author):

I thank the authors for their further clarifications and the revised manuscript. In my opinion, the computational methodology is now clear (save a minor issue, see below). Hence, I'm happy to recommend this revised manuscript for publication in Nature Communications.

As a minor suggestion, I should note that the explanation in Supplementary Note 5 is still not completely clear to me, especially what is meant by "completely relaxing angle parameters". The OPLS-AA force field associates with each angle θ two parameters: the mean angle θ_0 and the force constant K_θ . These are fixed values for the OPLS-AA force field that were determined by higher-level ab initio calculations. While a standard relaxation of the molecule using the OPLS-AA force field will change the angles θ , it will not change the force field parameters θ_0 and K_θ . So, if I understand the methodology of "completely relaxing angle parameters" correctly, the authors chose to put these force constants K_θ to zero. Is that correct? If so, I would rephrase the explanation in Supplementary Note 5 to make this clear.

Point-by-Point to Reviewers' Comments

Reviewer #3 (Remarks to the Author):

I thank the authors for their further clarifications and the revised manuscript. In my opinion, the computational methodology is now clear (save a minor issue, see below). Hence, I'm happy to recommend this revised manuscript for publication in Nature Communications.

As a minor suggestion, I should note that the explanation in Supplementary Note 5 is still not completely clear to me, especially what is meant by "completely relaxing angle parameters". The OPLS-AA force field associates with each angle θ two parameters: the mean angle θ_0 and the force constant K_θ . These are fixed values for the OPLS-AA force field that were determined by higher-level ab initio calculations. While a standard relaxation of the molecule using the OPLS-AA force field will change the angles θ , it will not change the force field parameters θ_0 and K_θ . So, if I understand the methodology of "completely relaxing angle parameters" correctly, the authors chose to put these force constants K_θ to zero. Is that correct? If so, I would rephrase the explanation in Supplementary Note 5 to make this clear.

Response: We thank the Reviewer for the suggestion. Yes, we relax the angle parameters by setting the force constants (k_θ) of all angles in n-hexane and cyclohexane to zero. A clarification note is included in Supplementary Note 5.